



# Analysis of recent lower stratospheric ozone trends in chemistry climate models

Simone Dietmüller[1], Hella Garny[1,2], Roland Eichinger[2,1], and William T. Ball[3,4,5]

[1]Deutsches Zentrum für Luft- und Raumfahrt (DLR), Institut für Physik der Atmosphäre, Oberpfaffenhofen, Germany
[2]Ludwig Maximilians Universität, Faculty of Physics, Institute for Meteorology, Munich, Germany
[3]Institute for Atmospheric and Climate Science, Swiss Federal Institute of Technology Zurich, Zurich, Switzerland
[4]Physikalisch-Meteorologisches Observatorium Davos World Radiation Centre, Dorfstrasse 33, 7260 Davos Dorf, Switzerland
[5] Delft University of Technology, Delft, The Netherlands

*Correspondence to:* S. Dietmüller (simone.dietmueller@dlr.de)

**Abstract.** Recent observations show a significant decrease of lower stratospheric (LS) ozone concentrations in tropical and mid-latitude regions since 1998. By analyzing 31 chemistry climate model (CCM) simulations performed for the Chemistry Climate Model Initiative (CCMI), we find a large spread in the 1998–2018 trend patterns between different CCMs and between different realizations performed with the same CCM. The latter, in particular, indicates that natural variability strongly

influences LS ozone trends. However none of the model simulations reproduces the observed ozone trend structure of coherent negative trends in the LS. In contrast to the observations, most models show a dipole trend pattern in the LS with negative trends in the tropics and positive trends in the northern mid-latitudes or vice versa. To investigate the influence of natural variability on the LS ozone trends we analyze the observational trends and the models' trend probability distributions for slightly varied post-ODS (ozone depleting substances) periods. Generally, modeled and observed LS trends remain robust for different

post-ODS periods, however observational data show a systematic change towards weaker mid-latitude trends forced by natural variability for certain periods. Moreover we can show that in the tropics the observed trends agree quite well with the models' trend distribution, whereas in the mid-latitudes the observational trend is a rather extreme value of the models' distribution. We further investigate the LS ozone trends for extended periods reaching into the future and find that all models develop a dipole trend pattern in the future, i.e. in almost all models the trends converge to constant values for the entire period 1998–2060.

An investigation of interannual ozone variability also reveals a clear dipole pattern in ozone variability in all CCMs and in observational data, however it is more pronounced in the models. Thus, although the LS ozone variability pattern is similar, the probability of overall negative LS ozone trends simultaneously in the tropics and mid-latitudes is higher in observations. To access the dynamical influence on LS ozone trends, the models' tropical upwelling trends are correlated against their LS ozone trends. For the period 1998–2018 tropical ozone trends are negatively correlated (-0.83) and mid-latitude trends are positively

correlated (+0.49). However, the correlation in the mid-latitudes is rather weak and not robust for slightly varied time periods, which indicates that other processes like two-way mixing play an important role here, too.



# 1   Introduction

Stratospheric ozone is essential for protecting the Earth's surface from ultra violet radiation, which is harmful for plants, animals and humans. Human-made ozone depleting substance (ODS) emissions significantly reduced ozone concentrations for some decades after 1960. After controlling the use of ODSs by the 1987 Montreal protocol, however, ODSs started to decline in the mid-to-late 1990s (e.g. Newman et al., 2007; Chipperfield et al., 2017). As a consequence, total stratospheric ozone is expected to recover in the future. Dhomse et al. (2018) have analyzed the recovery of stratospheric ozone mixing ratios of the CCMI-1 (Chemistry Climate Model Intercomparison project part 1) climate projection simulations. They found that the ozone layer is simulated to return to pre-ODS level between 2030 and 2060 depending on the region. However, they discovered a large spread among the individual models, which shows that there are many uncertainties in these projections. The evolution of stratospheric ozone in the 21st century does not only result from a decrease in ODSs but also from an interplay between changes in both the atmospheric composition and the circulation (World Meteorological Organization (WMO) 2014). Increasing anthropogenic greenhouse gas (GHG) emissions ($CO_2$, $CH_4$, $N_2O$) lead to enhanced tropical upwelling and thereby to an acceleration of tracer transport along the stratospheric overturning circulation (e.g. Butchart, 2014; Eichinger et al., 2019). On the other hand, increasing GHGs also slow down ozone depletion through GHG-induced stratospheric cooling (e.g. Jonsson et al., 2004; Oman et al., 2010; Bekki et al., 2013; Dietmüller et al., 2014; Marsh et al., 2016) and emissions of $CH_4$ and $N_2O$ additionally impact ozone by chemical processes (e.g. Ravishankara et al., 2009; Kirner et al., 2015; Revell et al., 2012; Winterstein et al., 2019).

In the recent years, a number of studies have analyzed observational records to identify ozone trends in the stratosphere (e.g. Harris et al., 2015; Steinbrecht et al., 2017; Weber et al., 2018). These studies consistently report an ozone recovery in the upper stratosphere after the turnaround of the ODS concentrations in the year 1998 (start of the post-ODS period). In the lower stratosphere (LS), however, most observed ozone trends are statistically not significant for that relatively short period due to large internal variability and instrumental difficulties (e.g. Steinbrecht et al., 2017). Subsequently, Ball et al. (2018) analyzed LS ozone trends from satellite data between 1998 and 2016 in detail making use of a dynamical (multiple) linear regression analysis. This way, they identified a statistically significant decline of LS ozone between 60°S and 60°N in that period, by approximately 2 DU in LS below 24 km of altitude. The implication was that the stratospheric ozone column was continuing to decline, because the LS ozone reduction more than offsets the positive trend in the upper stratosphere. Shortly afterwards Wargan et al. (2018) studied ozone trends in the reanalysis products MERRA-2 and GEOS-RPIT. In the tropics they detected a positive ozone trend in a 5 km layer above the tropopause and a negative trend at 7-15 km above the tropopause. Nevertheless, in the northern and southern mid-latitude LS they detected a negative ozone trend. As such, there are some similarities to the findings of Ball et al. (2018), but there are also major quantitative differences, for example the positive trend in the 5 km layer or a missing overall negative trend in the LS. Wargan et al. (2018) suggested that the negative mid-latitude trend might be explained by enhanced isentropic transport between the tropical and mid-latitude LS. However, the recent study of Orbe et al. (2020) explicitly demonstrated that in the NH this mid-latitude ozone decrease is primarily associated with large scale advec-



tion. By means of using a chemistry transport model (CTM) and extending the analysis period to the year 2017, Chipperfield et al. (2018) suggested that the negative LS ozone trends are only a result of large natural variability. They showed that there is a strong positive ozone anomaly in 2017 which is driven by short term dynamical transport of ozone, and concluded that this points to large year-to-year variability rather than to an ongoing downward trend. However, an update of the data set which

was used in Ball et al. (2018) showed that the large interannual variability alone cannot explain the entire trend in Chipperfield et al. (2018) (see Ball et al., 2019a): the larger year-to-year variability in the SH was implicated to results from a non-linear interaction between the quasi-biennal-oscillation (QBO) and seasonal variability and despite this large variability the observed negative LS ozone trend remains.

To get more confidence in future projections of the ozone layer it is important to know how well chemistry climate mod-

els (CCMs) simulate the observed post-ODS peak ozone trends. A direct comparison between the CCM multi-model-mean (MMM) values and observational data showed that the ozone trend profiles of modeled MMM data agree well with observations, except in the lowermost mid-latitude stratosphere (WMO, 2018; SPARC CCMVal, 2010). The most recent study of Ball et al. (2019b) investigated LS ozone trends of the 1998–2016 period in merged satellite data and compared them to the ozone trends in CCMs using the climate projection simulations of the CCMVal2 project. Similar to the observations, the

CCMs showed a decline in LS ozone in the tropics, likely due to enhanced tropical upwelling, following from an increase in greenhouse gases (see e.g. Randel et al., 2008). In contrast to the observations, however, models do not show a decrease, but rather an increase in LS mid-latitude ozone. Ball et al. (2019b) argue that these discrepancies in the LS between models and observations can possibly be explained by differences in the horizontal two-way mixing between the tropics and mid-latitudes, though they did not provide explicit evidence from the models, only observations (see also Wargan et al., 2018). This study

suggested that the negative mid-latitude observational trend is caused by an intensification of two-way mixing (by analyzing effective diffusivity in reanalysis data). On the other hand enhanced downwelling of ozone-rich air to the mid-latitudes could consequently lead to a positive trend in the mid-latitudes. Apparently, the processes that determine mid-latitude LS ozone in models and observations are not understood so far.

In the present study, we seek to quantify whether the observed LS ozone trends lie within the suite of modeled trends. If yes, this would imply that the observed trend is just one realization of possible trends given within the large year-to-year variability. If not, this would imply that either models do not represent year-to-year variability correctly, or that there is a forced trend in the real world that is not adequately represented in the models. In contrast to the study of Ball et al. (2019b) we are using

the simulation data of a more recent inter-model comparison project (namely the Chemistry Climate Model Initiative, phase 1, CCMI-1) and analyze the ozone trends for a wider range and for improved current state-of-the-art CCMs including all their ensemble simulations.

A brief description of the model simulations, of the observational data set and of the used statistical methods is presented in Section 2.1. In Section 3 we show our results: in section 3.1 we provide a detailed comparison of ozone trends over the years

1998–2018 in different CCM simulations and observations, with a focus on LS ozone trends and we investigate how natural





variability influences these LS ozone trends (Section 3.2 and 3.3). Moreover we link LS ozone trends with stratospheric upwelling trends (Section 3.4), and compare the year-to-year variability in observations and in all CCM simulations (Section 3.5). A discussion of the reasons for the disagreement in the LS mid-latitude ozone trend between models and observations and the conclusions follow in Sections 4 and 5, respectively.

# 2 Data and Methods

## 2.1 Models and Simulations

In the present study, we analyze the model output from overall 18 state-of-the-art CCMs from the Chemistry Climate Model Initiative phase 1 (CCMI-1, Morgenstern et al. (2017)). Tab. 1 lists all these CCMs together with their references, their under-

lying sea surface temperatures (SSTs), and the simulation type considered. A detailed overview of all models that participated in CCMI-1 can be found in Morgenstern et al. (2017). We mainly evaluate the long term free running simulations of CCMI-1 (REF-C2), as they span the time period we are interested in (namely 1998–2018). We do not use the the free running simulations of the recent past or the specified dynamics simulations, as they only span the period from 1998 to 2010. Moreover we want to point out that the specified dynamics simulations performed for CCMI do not represent stratospheric circulation

better than the free running simulations: Chrysanthou et al. (2019) compared stratospheric residual circulation among specified dynamic simulations and found that the spread in these simulations is even larger than in REF-C2. For the REF-C2 model simulations used in our study, all available ensemble members of the individual models are taken into account. The ensemble size of a certain simulation (if ensemble simulations were performed) is also given in Tab. 1 (brackets after simulations). Thus for the REF-C2 simulation 18 models performed a total of 31 realizations (6 models performed multiple ensemble members

simulations). The REF-C2 simulations include hindcast and forecast periods spanning 1960–2100. They are all free running simulations, thus each model simulation has its own internal variability. Note that REF-C2 simulations use a variety of different SSTs and SICs (sea ice concentrations), either prescribed climate model SST fields from offline model simulations (of the same or of a different model) or they are coupled to an interactive ocean and sea ice module. Moreover the representation of the QBO is different across the CCMs, with models having an internally generated QBO (e.g. MRI, EMAC-L90), nudged QBO (e.g.

NIES, WACCM, SOCOLv3, EMAC-L47, EMAC-L47-o) or no QBO (e.g. CMAM, LMDZ). REF-C2 reference simulations follow the WMO (2011) A1 scenario for ozone-depleting substances and the RCP 6.0 scenario (Meinshausen et al., 2011) for other greenhouse gases, tropospheric ozone precursors, and aerosol and aerosol precursor emissions. For anthropogenic emissions, the CCMI recommendation was to use MACCity (Granier et al., 2011) until 2000, followed by RCP 6.0 emissions. Besides the REF-C2 simulations we also consider the 11 sensitivity simulations with fixed greenhouse gases (fGHG) in our

analysis. This sensitivity scenario is based on the REF-C2 simulation, but the GHGs $CO_2$, $CH_4$, $N_2O$, and other non-ozone depleting GHGs are held at their 1960 value, thus we can study the impact due to ODS only (i.e in the absence of GHG induced climate change). All models providing this sensitivity simulations are given in Tab 1.





**Table 1.** Overview of the CCMI simulations, analyzed for the present study. For the individual CCMs their reference(s), their SSTs and their available simulations (REF-C2 and fGHG) are given. The numbers in brackets behind the simulations indicate the number of realizations of each REF-C2 and fGHG simulation. Detailed information about the models' SSTs and the models' representation of the QBO are given in the supplement of Morgenstern et al. (2017).

| CCMI Model | Reference(s) | SSTs | Analyzed Simulation |
|---|---|---|---|
| CMAM | Jonsson et al. (2004) | prescribed | REF-C2(1), fGHG(1) |
| | Scinocca et al. (2008) | | |
| CESM1-WACCM | Solomon et al. (2015); Garcia et al. (2017) | interactive | REF-C2(4)*, fGHG(1) |
| | Marsh et al. (2013) | | |
| EMAC-L90 | Jöckel et al. (2010, 2016) | prescribed | REF-C2(1), fGHG(1) |
| EMAC-L47 | Jöckel et al. (2010, 2016) | prescribed | REF-C2(1) |
| EMAC-L47-o | Jöckel et al. (2010, 2016) | interactive | REF-C2(1)** |
| GEOSCCM | Molod et al. (2012, 2015) | prescribed | REF-C2(1) |
| | Oman et al. (2011, 2013) | | |
| MRI | Deushi and Shibata (2011) | interactive | REF-C2(1) |
| | Yukimoto et al. (2011, 2012) | | |
| SOCOLv3 | Stenke et al. (2013); Revell et al. (2015) | prescribed | REF-C2(1) |
| NIWA-UKCA | Morgenstern et al. (2009, 2013) | interactive | REF-C2(5), fGHG(3) |
| | Stone et al. (2015) | | |
| ULAQ | Pitari et al. (2014) | prescribed | REF-C2(3), fGHG(1) |
| HadGEM | Walters et al. (2014); Madec et al. (2015) | interactive | REF-C2(1) |
| | Hunke et al. (2010); Morgenstern et al. (2009) | | |
| | O'Connor et al. (2014); Hardiman et al. (2017) | | |
| UMUKCA | Morgenstern et al. (2009); Bednarz et al. (2016) | prescribed | REF-C2(2) |
| ACCESS-CCM | Morgenstern et al. (2009, 2013) | prescribed | REF-C2(3), fGHG(1) |
| | Stone et al. (2015) | | |
| NIES | Imai et al. (2013); Akiyoshi et al. (2016) | prescribed | REF-C2(1), fGHG(1) |
| UMSLIMCAT | Tian and Chipperfield (2005) | prescribed | REF-C2(1), fGHG(1) |
| CHASER | Sudo and Akimoto (2007) | interactive | REF-C2(1), fGHG(1) |
| LMDz-REPROBUS | Marchand et al. (2012); Szopa et al. (2013) | interactive | REF-C2(1) |
| | Dufresne et al. (2013) | | |
| CESM1-CAM4-Chem | Tilmes et al. (2016) | interactive | REF-C2 (3) |

* The forth ensemble of WACCM (WACCM-4) was provided by M. Abalos; ** EMAC-L47 simulations are not ensembles, as one simulation is with prescribed SSTs and one with interactive ocean





## 2.2 Observational data

For observations, we make use the BAyeSian Integrated and Consolidated (BASIC) ozone composite that merges SWOOSH (Davis et al., 2016) and GOZCARDS (Froidevaux et al., 2015) through the BASIC method of Ball et al. (2017). The method was developed to account for artefacts in composite datasets that are a consequence of merging observations from different instruments that each have unique spatial and temporal observing characteristics. As a result, these artefacts can alias in regression analysis and bias, e.g., trend estimates (see examples in Ball et al. (2017)). BASIC composites aim to account for and reduce artefacts using an empirically driven Bayesian inference methodology, but it relies on the availability of already developed ozone composites. Here, $\text{BASIC}_{SG}$ has been extended to the end of 2019 using the latest versions of GOZCARDS, v2.20, and SWOOSH, v2.6. As such $\text{BASIC}_{SG}$ covers 1985-2019 as monthly mean zonal means on a $10°$ latitude grid from $60°$S–$60°$N and over a pressure range of 147–1 hPa ($\sim$13–48 km). $\text{BASIC}_{SG}$ was presented in Ball et al. (2018), and a sensitivity analysis of trends was applied to it in Ball et al. (2019a), with examples of data artefacts that it addresses in the accompanying appendix and supplementary materials, respectively.

## 2.3 Statistical Methods

In some parts of our analysis, and to make a robust comparison between multiple models and a single real-world realization, i.e. observations, we form probability distributions to estimate the combined probability of the ozone trends from all REF-C2 models. To do so, we calculate the linear trend and the associated uncertainty using a least squares method for every simulation. Then to build the probability distribution of the models trend, first one of the 18 CCMI models is randomly selected, assuming that the models are randomly uniform distributed. In case the selected CCM has ensemble members, one of these members is then also randomly chosen. In the next step, the trend estimate ($t^{M_{i,k}}$) of the specific randomly selected CCMI model $M_i$ with ensemble member $k$ is calculated by randomly choosing a ozone trend value from the trends associated and assumed normal distribution $\mathcal{N}$, which is based on the mean $\mu_{M_{i,k}}$ and standard deviation $\sigma_{M_{i,k}}$ of the simulations linear trend. Thus we can write the trend estimate of the selected model simulation as: $t^{M_{i,k}} = \mathcal{N}(\mu_{M_{i,k}}; \sigma_{M_{i,k}})$. In order to take into account the uncertainty of the single observational dataset ($\sigma_{obs}$), we also add to the calculated model trend estimate a random estimate of the observational noise by taking the observational standard deviation of the linear regression coefficient. We repeat the above described procedure 50 000 times. With that we have a large amount of model trends and can build up a robust probability density function (PDF) of the REF-C2 ozone trends. From these estimated PDFs we can then estimate the probability of a given trend relative to the models - in order to compare the model simulations. We derive a "probability of disagreement" between the observational and the modeled trend distribution by taking the central interval of the models' trend distribution with the observed trend value as threshold of this interval. To calculate this central interval we order the the 50 000 values from the REF-C2 trend distribution according to their probability values and then sum up the ordered probability values until the value of the observed trend is reached. This probability value indicates our estimate of whether the observations agree with the models, i.e. high probability values indicate that a disagreement between models and observations is less likely due to chance.





## 3 Results

### 3.1 Ozone trends over the period 1998–2018 in CCM simulations and observations

In this section we analyze the ozone trends of all free running CCMI-1 simulations (REF-C2), including all ensemble realizations of each model for the post-ODS period 1998–2018 together with the observational data, $BASIC_{SG}$. We chose the period

1998–2018 to be consistent with the observational trend estimate in the ozone recovering phase as presented by Ball et al. (2018). By using the REF-C2 simulations we include a wide spectrum of SST variability in the different CCMs, as they use either an interactive ocean or prescribed SSTs from a coupled ocean-atmosphere model simulation (see Tab.1). Ozone trends are calculated by simple linear regression (see section 2.3), using the monthly deseasonalized ozone time series. We refrain from excluding sources of variability such as QBO, ENSO (El Nino Southern Oscillation), solar cycle or volcanic eruptions in

the regression analysis to capture the full range of variability of ozone trends over the given period. Hence our trend estimates have to be interpreted as resulting from both forced trends (e.g., via GHG increases and ODS decreases) as well as from natural and internal climate variability. In the following we compare the so calculated ozone trend from the observational data to the trend estimate by using a dynamical linear modeling (DLM) approach, that does attempt to take natural sources of variability into account, as presented in Ball et al. (2018, 2019a).

The panels of Fig. 1 show a latitude-pressure cross-section of the post-ODS ozone trend for observations (first panel of Fig. 1) and all free running CCMI model simulations. Generally, the linear trend fit we perform on the $BASIC_{SD}$ data yields similar spatial patterns and magnitudes to those estimated in Ball et al. (2018) with the DLM approach (see their Fig. 1f). There are a few small differences, e.g., our linear trend fit results in larger positive trends in the upper stratosphere over the

southern tropics of $\sim 1\%$, a slightly less negative trend in the northern hemisphere middle stratosphere ($<1\%$), and consistently large and negative trends close to 100 hPa in the tropics as opposed to a smaller and insignificant trend at around $10°S$ and over 100-80 hPa in the DLM estimate as shown by Ball et al. (2019a). Most notably, linear trend calculations result in small positive trends (up to $\sim 3\%$) in the southern mid-latitude lower stratosphere, as opposed to overall negative but insignificant trends reported by Ball et al. (2019a) in that region. However, the comparison reveals that the overall magnitude and

trend pattern is also captured by the simple linear regression, i.e. it is not dependent on the exact method used to calculate the trends. Therefore, we proceed with using a linear fitting approach for the comparison between observations and CCMs, though the above caveats should be kept in mind when comparing with a full regression analysis using DLM (Ball et al., 2019a).

Overall, large inter-model variability of the trends derived from the individual REF-C2 simulations (including all ensemble

members) is revealed in Fig. 1. Nevertheless, a number of features can be identified that are consistent over most models and all their ensemble members. In the upper stratosphere (1-10 hPa) nearly all simulations consistently show an overall positive trend in ozone. This ozone increase can be explained by the decrease of ODSs (see e.g. WMO, 2018) and by a slow down in ozone destruction rates as the stratosphere cools from GHG increases (see e.g. Portmann and Solomon, 2007). This upper stratospheric ozone trend has been found for climate model simulations and for observational data in several studies before





(e.g. SPARC CCMVal, 2010; Harris et al., 2015; Steinbrecht et al., 2017; Ball et al., 2018; WMO, 2018; Ball et al., 2019b). However, in the lower stratosphere (30-100/150 hPa) we find a wide spread in the ozone trends among the CCM simulations over recent decades. Many REF-C2 simulations exhibit negative trends in the tropical LS, and they are comparable to the observational trend in magnitude and structure. In agreement with earlier studies (e.g. WMO, 2018), we will show in Section 3.4 that

this tropical ozone decrease is related to enhanced tropical upwelling in a warmer climate. However, there are also simulations showing a positive LS ozone trend in the tropics (e.g. GEOSCCM, SOCOLv3, NIWA-1, WACCM-3, CAM4-1, LMDZrepro; note that the number of the ensemble run is denoted with -1, -2 and so on). At northern and southern mid- and high-latitudes most CCMs exhibit a positive trend, but with a pronounced spread between model simulations. Only a few simulations show negative trends in either northern or southern mid-latitudes (e.g. GEOSCCM, WACCM-3, WACCM-4), but it is important to

point out here that none of the 31 simulations reproduces the observed negative ozone trend pattern with ozone decrease covering the tropical belt and extending to the mid-latitude (50°S-50°N), as shown in the upper-left panel and previously in Ball et al. (2018, 2019a). This discrepancy in the LS ozone trend between observations and models has been reported before (e.g. ozone profile trends, based on CCMI simulations, (WMO, 2018), and in comparison to CCMVal-2 simulations (Ball et al., 2019b)). For CCMs that provide multiple ensemble members (WACCM, NIWA, ULAQ, ACCESS, CAM4 and UMUKCA),

we also identify a large spread in the simulated LS ozone trends. For example in WACCM two ensemble members simulate positive tropical ozone trends, while the two other members simulate negative tropical ozone trends. In WACCM (as well as in NIWA and CAM4), the coupled ocean allows for differences in the SST variability between the ensemble members, possibly explaining the large spread in tropical ozone trends. However, as is also the case for models with prescribed SSTs (ACCESS, ULAQ, UMUKCA), that exhibit a large spread between the simulations, the SST variability is not the only reason for the

different trend pattern, as was similarly reported and discussed by Ball et al. (2019b) for CCMVal-2 models. The large spread in LS ozone trends between ensemble members is further in agreement with the study of Stone et al. (2018). They used a nine member ensemble of a free running CCM simulation and showed that LS ozone trends over the years 1998–2016 are characterized by large internal variability, with e.g. the LS ozone trend ranging from +6% to -6% per decade. But note, again, that none of these ensemble members showed the coherent decrease in ozone in the tropics and extratropics as found in observations

(Ball et al., 2019b).

Following this qualitative discussion on the spread in the ozone trend pattern between the CCM simulations, we now turn to the LS ozone trends with a more quantitative comparison of the apparent inconsistencies between observations and CCMs. We calculate the trends of the deseasonalized LS ozone columns for the post-ODS period 1998–2018 in two regions: the inner

tropics (20°N-20°S) and in the northern mid-latitudes (30°N-50°N). We choose the northern mid-latitude band 30°N-50°N in order to be comparable to the study of Ball et al. (2019b). The pressure range of the lower stratosphere was taken to be 30–100 hPa for the tropics and 30–150 hPa for the mid-latitudes, to take into account the differences in the latitudinal tropopause heights. Trends and their uncertainties (represented by the 90% confidence interval of the linear slope) are shown for each of the 31 available REF-C2 simulations of 18 different CCMs in Fig. 2. We decided to focus on the northern mid-latitudes here,





**Figure 1.** Latitude-pressure cross section of the ozone trend over the post-ODS period 1998–2018 for the observational data set BASIC$_{SG}$ and for all CCMI REF-C2 simulations. Trends are given as relative ozone changes over the whole time period. Boxes illustrate the regions selected to integrate ozone in the LS for trend comparisons later in this study, i.e. in the tropics (20°N-20°S, 30–100hPa) and in the northern mid-latitudes (30°N-50°N, 30–150 hPa).



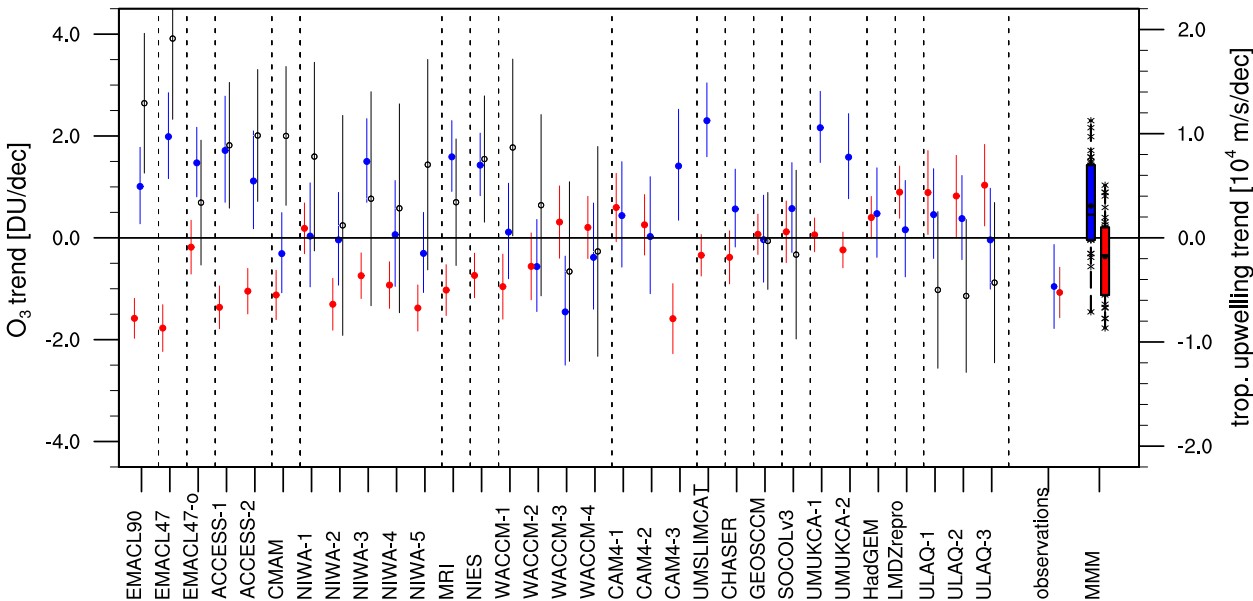

**Figure 2.** LS ozone trends and their uncertainties in the tropics (20°N–20°S, red dots) and northern mid-latitudes (30°N-50°N, blue dots) together with tropical upwelling trend (black circles, for all simulations providing TEM diagnostics) for the period 1998–2018 for all REF-C2 simulations. Dashed lines separate the individual models. Moreover, observational trends (1998–2018) and multi-model mean trends are given. Observational data are taken from BASIC$_{SG}$. Error bars associated with each LS ozone trend represent the 90% confidence intervals. The multi-model mean trends are shown as boxplots: the black solid line in the box indicates the median, the black point the MMM and the colored box ranges from the 25th to the 75th percentile of the trends. Crosses denote trends of individual model simulations not lying within the box.

because the SH mid-latitude trends are likely strongly influenced by the large chemical depletion of ozone within the polar vortex. However, we will come back to the LS ozone trends of the southern mid-latitudes later on (see section 4).

In the tropics about half (42%) of the REF-C2 simulations show a significant decrease, about the same (42%) show a non-significant change, and about 15% a significant increase in the integrated tropical LS ozone column. Note that significance is defined as the non-overlap of the error bars (90% confidence interval) with the zero trend. The resulting MMM ozone trend (see red bar on right of Fig. 2) is negative (-0.37 DU/dec), but it is insignificant due to the considerable spread among the different models, with the 25th-75th quantile of the distribution ranging from -1.12 to 0.20 DU/dec (see edges of box on right of Fig. 2). Note that for the MMM trend each of the 31 simulations is weighted equally, i.e. not taking into account that some models have multiple ensemble members, because the trend variation among ensemble members are as large as among the different





models. The observed tropical LS ozone trend of -1.07 DU/dec is statistically significant at the 90% level. Thus the observed tropical trend is more strongly negative than the MMM trend, but lies within the 90% confidence interval of the MMM trend (being [-1.76 DU/dec; 1.03 DU/dec]).

In the northern mid-latitudes less than half (40%) of the REF-C2 simulations show an increase in the LS ozone column, while
5   the remaining 60% of the simulations show a non-significant change (either positive or negative). There is only one simulation (WACCM-3) that shows a significant decrease in the mid-latitude LS ozone column, although it has a corresponding (non-significant) tropical trend. The resulting MMM trend in the northern mid-latitudes is positive (+0.63 DU/dec) with a high inter-model spread: the 25th to 75th quantile of the distribution ranges from -0.04 to 1.42 DU/dec. Note here, that the observational trend (-0.96 DU/dec) lies just outside the 90% confidence interval of the MMM trend in the mid-latitudes [-0.91 DU/dec; 2.16
10  DU/dec].

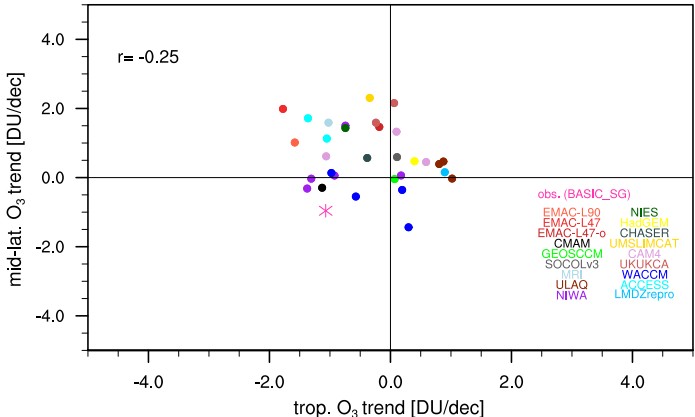

**Figure 3.** Inter-model correlation between tropical (20°S-20°N) and northern mid-latitude (30°N-50°N) LS ozone column trends, calculated over the period 1998–2018 for 31 CCMI REF-C2 simulations. All ensemble members of a particular model are shown in the same color. The observational ozone trends (BASIC$_{SG}$) are also included in here as a star.

.

Fig. 2 also reveals that over the years 1998-2018 more than half of the model simulations have a dipole trend pattern in the LS ozone column, i.e. the sign of the tropical ozone trend is opposite to that in mid-latitudes. A dipole trend pattern with negative LS ozone trend in the tropics and positive LS ozone trend in the northern mid-latitudes can be found for almost half the simulations (45%), and a dipole trend pattern with positive ozone trend in the tropics and negative trend in the northern
15  mid-latitudes is found in 13% of the simulations. The remaining simulations do not show this dipole, but either both have a positive trend in the tropics and the mid-latitudes (29%), or a negative trend in both tropics and mid-latitudes (13%, i.e. 3 simulations, namely NIWA-5, CMAM, WACCM-2). Only 3 out of 31 simulations simulate negative, but not significant trends both in the tropics and northern extratropics, thus they show a similar behavior to observations (see right of Fig. 2 and Ball et al.





(2019a)). However, the zonal trend patterns (see Fig. 1) reveal, that none of these three simulations reproduces the observed trend pattern with consistent negative trends from 50°S-50°N in the LS. Consequently it is important to keep in mind that the results of these (averaged) trends are depending on the choice of the latitude-pressure box, as the integration over a wider latitude band can lead to a cancellation of opposing trends.

To analyze whether a systematic relationship between the tropical and mid-latitude trend can be diagnosed from the CCM simulations, the simulated northern mid-latitude LS ozone trends are plotted against the simulated tropical LS ozone trends over the time period 1998-2018 for all 31 REF-C2 simulations and for the observed data-set BASIC$_{SG}$ in Fig. 3. As discussed above, in the LS the majority of the models have a negative ozone trend in the tropics and a positive trend in the northern mid-latitudes. This illustration again highlights that the trends estimated from observational data are lying on the outer edge of

the distribution of the modeled trends. The inter-model correlation between the tropical to mid-latitude trends is negative with a low not significant correlation coefficient (-0.25). Thus, for the chosen period the tropical ozone trends are significantly, but weakly linked to mid-latitude ozone trends in the models. We expect a negative correlation here, because increased tropical upwelling, leading to decreased tropical ozone, should be linked to increased mid-latitude downwelling, that would enhance ozone in mid-latitudes. The relationship of the trends to tropical upwelling will be further investigated in Sec. 3.4.

Overall we can conclude from the analysis of ozone trends in the suite of CCMI models (see Fig. 1, 2 and 3), that the LS ozone trends exhibit a considerably large spread across both the different models, but also across individual ensemble members, in particular in the mid-latitudes. This indicates that ozone variability considerably influences the LS trends, in agreement with the recent studies by Chipperfield et al. (2018) and Stone et al. (2018). However, even when considering the high variabil-

ity of possible trends in CCM simulations, the observational trends emerges as a rather unlikely realization of the simulations over the period 1998–2018. In the next section, we will analyze the robustness of this finding by varying the period of the trend calculation, and providing an in-depth statistical analysis of the likelihood of the observed trend lying within the suite of modeled trends.

## 3.2 Robustness of lower stratospheric ozone trends

In the previous section we found that the observed LS negative ozone trend in the mid-latitudes together with a simultaneous negative trend in the tropics is unlikely given the suite of CCM simulations. To further establish the robustness of this result, we here test whether this also holds for time periods that are slightly different to the post-ODS period 1998–2018 we considered before. Thus, in this section we first want to investigate how variability influences the ozone trends, and second we want to quantify the likelihood of the observed trend being a realization of the distribution of the modeled trends. To answer those

questions, we calculate the LS ozone trends by varying the start and end years of the time period. In Fig. 4 (a) and (b), the observed tropical and mid-latitude ozone trend in the LS is shown for start years varying from 1995–2001 (y-axes) and end years from 2013–2019 (x-axes). Both tropical as well as mid-latitudes LS ozone trends are consistently negative for all chosen periods in the observations (top raw). This is in line with the results of Ball et al. (2019a), who found that the observed negative sign of the tropical and mid-latitude trends remain insensitive to changing the end year. In the tropics, observational LS ozone





trends are consistently negative with values between -0.64 and -1.24 DU/dec for all possible start end year combinations. In the mid-latitudes the trends are also negative for all shown time periods, but are more variable than in the tropics (values range between -0.11 and -1.22 DU/dec). In particular at mid-latitudes, the strongest negative trends are found for start years of 1996 to 1998, and a sudden decrease in the trend magnitude is found for the start year 1999 and 2000. Thus, the analysis in Ball

et al. (2018, 2019a) and in the preceding section focused on a period with particular strong negative mid-latitude ozone trends. Possible reasons for the sudden change in the trend, like the strong ENSO event in 1998, are discussed in Sec. 4. Note that the trend magnitude goes back up again for the start year 2001, so it again looks like interannual variability influences the observational mid-latitude trends.

Fig. 4 (c) and (d) display the tropical and mid-latitude trends as a function of start and end year derived from the model simulations. To do so, a robust estimate of the trend probability distribution considering all model simulations was derived (see Section 2.3) and from this distribution the most likely trend is shown (see peak in the models' trend probability distributions of Fig. S1 and S2 in the supplement). In the tropics the ozone trends derived from the REF-C2 simulations are negative and, range from -0.74 to +0.02 DU/dec. In the mid-latitudes the trends are positive for all possible start/end year combinations, with

values ranging from +0.4 to +1.48 DU/dec. In contrast to the sudden change in the mid-latitude observational trend for start years 1999 and 2000, in the REF-C2 simulations no such systematic change can be found. The estimated probability distributions of the trends from the REF-C2 simulations (see Figs. S1 and S2 in the supplement) are typically symmetric around their maximum value and show a single, central peak. The width of the distribution changes for varying start/end year combination, with narrower distributions for longer time periods. Moreover, the visual inspection of the distribution implies the tropics

(Fig. S1) have generally Gaussian-like distributions, whereas the mid-latitudes (Fig. S2) often show a more peaked structure, i.e. with heavier trails. Nevertheless, as an estimate of the width of the models trend distribution, we show in Fig. 4 (e) and (f) the standard deviation of the models distribution (in DU/dec) in the tropics and mid-latitudes, respectively. For longer time periods (values in lower right corner) the standard deviation of the models' trend is smaller, i.e. the distribution is narrower. This indicates that the influence of natural variability is less important for longer time periods, as should be expected.

Given the distributions representing the combined trends of the models, we can now quantify the disagreement between the observational trend estimate and the models' trend probability distributions for each start/end year combination. In Figs. 4 (g) and (h) the "probability of the disagreement" between observational and modeled LS ozone trends is given for the tropics and the mid-latitudes. The value of the "probability of disagreement" is calculated by the central interval of the models' probability

distribution when taking the observed trend value as threshold of this interval. Thus, a probability value of 90% indicates that the observed trend falls within the inner 90% of the distribution, i.e., only 10% of the distribution is more extreme than the observed trend: the smaller the given "probability of disagreement" value, the higher is the probability that the observed trend lies within the models' distribution. In the tropics, the observed LS ozone trend falls within the 13% to 73% interval of the modeled probability distribution, i.e. the observed trends are generally likely representations of the models' trends. The agree-

ment is best for short time periods (values in diagonal in Fig. 4 (g)), mostly because of the broader distribution (see Fig. 4 (e)





and Fig. S1). Also for early start years (in particular 1995) and end years ranging from 2013 to 2018, the disagreement is small, because model trends are strongly negative for this period (see Fig. 4 (c)). In the mid-latitudes, the observed trend generally lies at more distant parts of the models' trends distribution (73% to 96%), i.e. the observed trend is a more extreme value in the models' distribution. The disagreement is smallest for both the earlier periods (lower left, start years 1995–1997 and end

years 2013–2015) and the later periods (upper left, start years 1999–2001 and end years 2017–2019). This coincides with the generally smaller negative trends in those periods in observations (see Fig. 4 (b)) and rather constant trend distributions in the models (see Fig. 4 (d)). For the periods with the strongest negative observed trend (start years 1996–1998), the observed trend lies within the central 90% or higher of the models' distribution, i.e. is an unlikely representation from the modeled trends. The sudden decrease in the observed trend magnitude for start years 1999 (Fig. 4 (b)) is reflected by a decrease of the central

interval to about 75%. In general, one might have expected that longer periods lead to better agreement of the observed and modeled trend due to the smaller influence of variability (see Figs 4 (e) and (f))- as we do in the models- however, we do not find this to be true for either the tropics or the mid-latitudes.

   While the above analysis quantified the "probability of disagreement" separately for the tropical and mid-latitude LS ozone

trends, we now turn to the question how the trends in those two regions are related. In Fig. 3 we showed for the period 1998–2018, that the majority of the models have a dipole trend structure in LS ozone and that tropical and mid-latitude trends are (weakly) negatively correlated (r=-0.25). Tab. 2 now quantifies the inter-model correlation (based on all 31 REF-C2 simulations) between tropical and mid-latitude LS ozone trends together with the percentage of the simulations with a specific sign structure in the LS trends (i.e. positive or negative trend in tropics / mid-latitudes) for four different post-ODS periods, i.e.

1998-2018, 1999-2019, 2000-2020 and 2001-2021. With that we want to get an impression of the results robustness to slightly different post-ODS periods of the same length. For all given periods, the fraction of models exhibiting a dipole trend pattern with a negative ozone trend in the tropics and a positive trend in the northern mid-latitudes is highest. Depending on the time period, overall about 50 to 60% of the 31 model simulations show a dipole trend pattern (either with negative trend in the tropics and positive trend in mid-latitudes, or vice versa). Up to 32% of the simulations, exhibit the observational trend pattern with

negative tropical and negative mid-latitude LS ozone trend, but these are mainly specific ensemble member simulations of two models, NIWA and WACCM. The correlation coefficients between tropical and mid-latitude ozone trends for the different periods reveal that the weak negative correlation found for the 1998–2018 period is not robust, as correlations are essentially zero for other chosen periods. Thus, while we would expect opposing effects of tropical upwelling and mid-latitude downwelling trends on the ozone trends, we do not find robust relationships between ozone trends in the tropics and mid-latitudes. However,

as discussed for the period 1998–2018 in Sec. 3.1, coherent negative ozone trends throughout the tropics and mid-latitudes are not found in the model simulations even when trends are diagnosed as negative in the tropical and mid-latitude boxes. Thus, the lack of relation of tropical to mid-latitude ozone trends found above might also be a result of the fixed definition of the boxes, as will be discussed further in Sec. 4.





**Table 2.** Inter-model correlation coefficient between tropical and mid-latitude LS ozone trends and percentage of the 31 REF-C2 simulations having the specific tropical/mid-latitude trend pattern of negative/positive, positive/positive, positive/negative or negative/negative sign. All calculations are given for the different post-ODS periods 1998-2018, 1999-2019, 2000-2020 and 2001-2021. Note that all correlation coefficients are not significant at 90% level.

|           | r     | neg./pos. | pos./pos. | pos./neg. | neg./neg. |
|-----------|-------|-----------|-----------|-----------|-----------|
| 1998-2018 | -0.24 | 45%       | 29%       | 13%       | 13%       |
| 1999-2019 | -0.15 | 48%       | 23%       | 6%        | 23%       |
| 2000-2020 | -0.05 | 39%       | 19 %      | 10 %      | 32%       |
| 2001-2021 | -0.02 | 42%       | 19%       | 12%       | 26 %      |

### 3.3 Convergence of future lower stratospheric ozone trends

In the last section, the robustness of ozone trend was analyzed for time periods of up to 25 years. We will show in the following, as the considered time periods become longer. the influence of natural variability decreases more and more, and the trends converge to the trend forced by long-term trends in GHG and ODS concentrations. To analyze the timing and the values of the

trends' convergence, we extend the period for the trend calculation into the future for all REF-C2 simulations.

Fig. 5 shows the tropical and northern mid-latitude LS ozone trends together with the tropical upwelling trend (black; if available) for periods with the fixed start year 1998, and the end year varying from 2013 up to 2060, by extending the time period by steps of one year. For reference, the observational trends are shown in the upper left panel of Fig. 5, with the last available end point in the year 2019. As shown in the last section, the trends derived from observational data are consistently

negative both in the tropics and in the northern mid-latitudes.

As discussed in Sec. 3.1, the ozone trends exhibit strong inter-model spread for the observational time periods. Both tropical and mid-latitude ozone trends in the individual model simulations vary considerably for different end point years within the observational period (left of the vertical dashed gray lines), with the northern mid-latitude trend more variable than the tropical trend. For longer time periods extending into the future, for all simulations the uncertainties in the LS ozone trends decline and

trends converge. All model simulations consistently evolve a dipole trend pattern, with significant and persistent negative trends in the tropics and persistent positive trends in the northern mid-latitudes (for a summary, see also Fig. 8). However the timing of convergence of the trends to the dipole trend pattern is rather different in the simulations, as can be inferred from Fig. 5, i.e. the convergence appears to be model dependent. For some models, the trends vary little for end years after 2020 (e.g. MRI in Fig. 5), while in other models, the trends still vary considerably until end years around 2030 to 2040 (e.g. the four WACCM

ensemble members in Fig. 5). The timing of the convergence basically is controlled by the ratio of the year-to-year variability to the strength of the forced trends. The relative forcing by ODS versus GHG changes over time for the given periods, and thereby the forced ozone trends vary over the time periods, making it difficult to quantify an exact date of convergence. Still, the trend estimates for the entire period 1998 to 2060 do converge to stable values for almost all models, thus representing the forced trend for this time period. For all model simulations, the long-term LS ozone trends over the entire period (1998–2060)





are negative in the tropics and positive in the extratropics, but the trend magnitudes vary between the models. Thus model trends vary from -0.10 to -1.32 DU/dec in the tropics, and from +0.39 to +2.00 DU/dec in the mid-latitudes. Comparing this to the model range of the somewhat shorter time period 1998–2040, we see that the tropical trend (+0.06 to -1.12 DU/dec) has not converged to the end point values of 2060, yet. The mid-latitude trend (+0.54 to +2.15 DU/dec) is however close to the 2060 values.

### 3.4 Influence of tropical upwelling on LS ozone trends in CCMs

In the preceding sections, the analysis focused on quantifying the LS ozone trends for the suite of CCMI-1 model simulations. As is well known from earlier studies, tropical upwelling significantly influences stratospheric ozone in the tropics (e.g. Oman et al., 2010). Enhanced tropical upwelling leads to more transport of tropospheric ozone-poor air to the tropical LS, and moreover to faster removal of ozone in the tropical pipe reduces the residence time in the LS. In Fig. 2, the tropical upwelling trends at 70 hPa for the latitude band between 20°N-20°S are given for all CCM simulations that provide residual circulation diagnostics. Note that we calculate tropical upwelling via the TEM meridional velocity (v*) field, because of inconsistencies in the archived TEM vertical velocity fields (w*), as model groups calculated w* slightly differently (see supplement of Dietmüller et al. (2018), Chrysanthou et al. (2019) and Eichinger and Šácha (2020)). In general, we find that models with strong positive upwelling trends show strongly negative tropical ozone trends, and often positive mid-latitude trends.

To analyze this more quantitatively, we calculate the inter-model correlation between tropical upwelling and LS ozone column trends for four different post-ODS periods of the same length (1998-2018, 1999-2019, 2000-2020, 2001-2021), presented in Fig. 6 (a) and (b). Trends are shown for all REF-C2 simulations that provided both TEM diagnostics and ozone fields (in total 22 REF-C2 simulations of 11 different models). We use multiple periods for calculating the correlations to test the robustness of the correlation. Year-to-year variability indeed causes trends to differ between the periods considered, however, the periods can of course not be considered as unrelated due to the large overlap.

In Fig. 6 (a), the tropical upwelling trend at 70 hPa is plotted against the tropical LS ozone column (20°N-20°S, 30-100 hPa) trend across the 22 model simulations. As mentioned before, most model simulations show either an increase in tropical upwelling with a decrease in tropical LS ozone, or a decrease in tropical upwelling with an increase in LS ozone. Again we find a large spread in the trends of both quantities among the models, but the spread is just as strong among the ensemble member simulations (ensemble members are indicated by the same color). For example the individual WACCM ensemble members (blue dots) exhibit a spread about half of the total inter-model spread. The negative correlation between the tropical upwelling trend at 70 hPa and the tropical ozone column trend is high and significant for all analyzed post-ODS periods (r=-0.83, -0.78, -0.71, -0.82). This is in line with previous studies (e.g. Oman et al., 2010; SPARC CCMVal, 2010). On the right side of Fig. 6 (a), the profiles of the inter-model correlations are given: the tropical LS ozone column trends are correlated against the tropical upwelling trends at 10, 30, 50, 70 and 100 hPa. The correlation values between LS tropical ozone trends and upwelling trends are highest for levels between 30 and 70 hPa.

Tropical upwelling has to be balanced by extratropical downwelling, which transports ozone-rich air downward. Thus, a positive correlation of tropical upwelling to extra-tropical ozone could be expected. In Fig. 6 (b) the tropical upwelling trend at



hPa is plotted against the northern mid-latitude LS ozone column (30°N-50°N, 30-150 hPa) trend. As expected, the correlation between those quantities is positive, but weaker than the relation of upwelling trends to tropical ozone trends. As shown in the right panel of Fig. 6 (b), correlations between mid-latitude ozone to upwelling are larger at 30 hPa when considering all four time periods. This might be due to the fact that tropical upwelling at larger pressure level matches the downwelling branch

better, as the effect of the meridional transport from tropics to mid-latitudes via the shallow branch is excluded. Compared to the inter-model correlation with tropical ozone the correlation with the northern mid-latitude ozone column trend is weaker and positive for the slightly different post-ODS periods (r=0.49, 0.47, 0.24, 0.28). Note that the correlation coefficients of the last two periods are not significant at the 95% level. As noted before, this positive correlation can be explained by the fact that enhanced downwelling brings ozone rich air from high altitudes to the LS mid-latitudes. The fact that the correlations

of mid-latitude ozone trends to tropical upwelling / downwelling trends are weaker compared to tropical ozone suggests, that mid-latitude ozone is not controlled only by vertical advection. Other transport processes are also relevant for mid-latitude ozone trends, for example meridional advection and mixing processes. A detailed discussion on this issue will follow in the Section 4.

### 3.5 Inter-annual variability of ozone

While a dipole pattern in tropical versus mid-latitude ozone trends was clearly found for long-term (forced) trends (i.e. for 1998-2060), in the following we will investigate whether this dipole pattern is also found for year-to-year variability. From this we aim to get a better understanding of how internal ozone variability influences the LS ozone trends, and whether variability in tropical and mid-latitude ozone are related. We present in Fig. 7 the interannual correlation coefficients between the tropical LS ozone column (20°S-20°N, 30-100 hPa) and the local ozone concentration at all latitudes and levels. We display this correlation

for the detrended annual mean ozone data over the period 1998-2018 for observations and over the period 1998-2060 for all individual model simulations. Of course the time period for the observational interannual correlation is much shorter, but interannual correlations of the model simulations are also robust for shorter periods (not shown here). Correlations are positive and close to one within the tropical LS for all models and observations. The positive correlation mostly peaks between 60 and 30 hPa, as the tropical ozone column is dominated by high ozone concentrations at the top of the box. However, there are

differences in the spatial correlation pattern within the tropics: for the observations as well as for some models (e.g. ULAQ, LMDZrepro), positive correlations are rather confined within the defined tropical box. Further, in observations and a number of models (e.g., EMAC, GEOSCCM, HadGEM, UMUKCA, NIES), the high positive correlations clearly peak between 30 and 50 hPa, while in other models (e.g., CMAM, NIWA) strong correlations of above 0.6 extend down to about 150 hPa. These structural differences are possibly caused by the models' different strengths in tropical upwelling or differences in the structure

of the tropopause layer.

Negative correlations to the tropical LS ozone column are evident for the observations and all models in the LS mid-to-high latitudes, mostly peaking around 40°N-50°N. The height of the maximal negative correlation varies from the upper boundary of the box around 30 to 50 hPa (e.g. in observations, ULAQ, HadGEM) to the lower boundary of the box around 100 to 150 hPa (e.g., MRI, CMAM, ACCESS), and in some models the maximum negative correlations extend through a deep layer





(e.g. SOCOLv3, NIWA). While the zonal structure of the interannual correlation patterns in both observations and models consistently shows a dipole pattern in ozone variability, differences in the correlation magnitude exist, e.g. the mid-latitude anti-correlation is more pronounced in the models. The correlation coefficients of the whole tropical LS ozone column to the mid-latitude LS ozone column is given in Tab. 3 (right column). The correlations are negative for all model simulations, and
significantly different from zero for all but two simulations (NIES and CMAM). The multi-model mean of the correlation coefficients is -0.49. The correlation of tropical to extratropical LS ozone derived from the observational data is only -0.21, and not significantly different from zero.

    The consistent dipole pattern in ozone variability in observations and models is likely the result of residual circulation variability. As discussed for the relation of ozone trends to upwelling trends, since enhanced tropical upwelling has to be
balanced by enhanced downwelling, low ozone values in the tropics are linked to high ozone values in the mid-latitudes and vice versa. To support this hypothesis, Tab. 3 provides the interannual correlation and their 95% confidence interval between tropical upwelling (w* at 70 hPa) and the tropical and the northern mid-latitude LS ozone column for the REF-C2 simulations. However we want to note here that the impact of downwelling on ozone trends is difficult to analyze, because of the different downwelling strengths in models and the different regions of downwelling. There is a large model spread in the interannual
variability correlation coefficients between downwelling (here tropical upwelling at 70 hPa is used as proxy - as it shows the best correlations in most of the models) and LS mid-latitude ozone (Tab. 3), showing the different influences of downwelling in the individual models. We suppose that downwelling is more pronounced in some models, or that the relative strength of downwelling to mixing is different in the models. Moreover simple detrending does not eliminate all long-term effects on this interannual processes. Consequently, tropical ozone trends are strongly driven by tropical upwelling, whereas mid-latitude
ozone trends are only partly controlled by mid-latitude downwelling processes, thus other processes may play a role there. Note further that the REF-C2 simulations have model-dependent height where the interannual correlations are best - for the tropics the models' best correlation height varies from 50 to 80 hPa and for the mid-latitudes it varies from 30 to 80 hPa (see Fig. S3 in the supplement). Thus, if e.g. the tropical ozone column in one individual model is determined by the variability at 30 hPa, the correlation at 30 hPa is best. In another individual model the ozone column is determined by the variability of the
entire box, and thus the correlation in a lower altitude should be better. As consequence we have to be careful with comparing these correlations.

    From Tab. 3 we see that the residual circulation (in terms of tropical upwelling) strongly impacts LS ozone variability in the models, in particular in the tropics. For most of the individual model simulations, the correlation coefficients of tropical upwelling to tropical LS ozone are negative (except for ULAQ), mostly with rather high values (MMM -0.56±0.19). Mid-
latitude LS ozone correlates positively with tropical upwelling in all models, but with generally weaker correlation coefficients (MMM 0.31±0.18). The model spread in the correlation value is large.

    Thus, overall we find similar behavior to what has been shown for the intermodel correlations of the trend (compare with Fig. 6). While in general tropical ozone anomalies (trends) are closely related to tropical upwelling anomalies (trends), and mid-latitude ozone anomalies (and less so trends) are anti-correlated to tropical ozone anomalies (trends), the mid-latitude
ozone anomalies (trends) are only weakly controlled by tropical upwelling anomalies (trends). Both observations and all indi-





vidual models show a dipole pattern in the LS ozone variability, thus indicating that processes driving variability are similar in observations and models. However, the dipole in the variability (i.e. relation of tropical to mid-latitude ozone) is less pronounced in observations compared to the model simulations. This could be due to the shorter time period, in particular given that the correlations estimated from the models display a large confidence interval despite the much longer time period they are

based on. On the other hand, the weaker "dipole"-like variability in observations could also indicate that the overall negative LS trend pattern is more likely in observations.

## 4 Discussion

In the previous sections we analyzed the ozone trends of recent post-ODS periods (i.e. 1998-2018 and variations on this period)

in detail and found that modeled and observational ozone trends disagree especially in the northern mid-latitude LS. One possible reason for the disagreement between modeled and observed LS ozone trends could be problems with the satellite records, as e.g. instrument biases and drifts can lead to large uncertainties in the observations (see e.g. Harris et al., 2015; Ball et al., 2017; Gaudel et al., 2018). However, assuming the observational data are correct, the question that arises from our results is whether the disagreement stems from the influence of natural variability, or whether it is related to the forced trend. In the

following, we will discuss this question using Fig. 8, which summarizes our results: it shows the observational as well as the REF-C2 MMM ozone trends and its spread for periods of different length, for the tropics, the northern mid-latitudes and the southern mid-latitudes in the upper and lower stratosphere. Note that for the calculation of the MMM trend in Fig. 8 only 10 model simulations are taken into account, as we directly compare to the MMM of fGHG simulations later on, which have a smaller sample size (see Tab. 1). With that we ensure that both simulations (fGHG and REF-C2) include the same models for

the MMM trend estimate. Moreover we exclude ULAQ for the MMM calculation, as it is an extreme values compared to other models, such that it has big influence on the MMM.

Beginning with the upper stratosphere, we see that the MMM ozone trends are positive and of the same magnitude in tropical and mid-latitude regions (Fig. 8a). This is in line with observational data (dots in Fig. 8a) and also all individual REF-

C2 simulations consistently show this feature (see Fig. 8a and Fig. 1). Note however, that the observational upper stratospheric ozone trend lies on the lower end of the modeled ozone trend range. The positive upper stratospheric MMM trend can be explained by the combined effect of still decreasing ODS concentrations at the beginning of the post-ODS trend period and by rising GHG concentrations causing stratospheric cooling. The contribution of the two effects is quantified by comparing the fGHG simulations to the REF-C2 simulations. In the former, the GHG-driven increase of the stratospheric circulation resulting

mostly from the increase in SSTs, as well as GHG induced stratospheric cooling is excluded. The upper stratospheric ozone trends in the fGHG simulations are positive for all analyzed periods, but considerably lower then the REF-C2 ozone trends, in particular for the later periods. The weaker trend in the fGHG simulations in upper stratospheric ozone can be explained by the missing additional ozone trend due to GHG induced stratospheric cooling (as ozone is photochemically controlled in





**Table 3.** Interannual correlation coefficients calculated between tropical upwelling (w* at 70 hPa, 20°S-20°N) and tropical LS ozone column (30-100 hPa, 20°S-20°N), as well as tropical upwelling (w* at 70 hPa, 20°S-20°N) and mid-latitude LS ozone column (30-150 hPa, 30°N-50°N). Correlation coefficients are based on detrended annual mean over the period 1998-2060. The 95% confidence intervals of the correlation coefficients is given in brackets, and non-significant correlation values are highlighted in gray. Moreover correlation values of observational data and the MMM correlation (mean over all model simulations, without ULAQ) is given.

| | $corr_{w*(70hPa),O_3(trop)}$ | $corr_{w*(70hPa),O_3(mid)}$ | $corr_{O_3(trop),O_3(mid)}$ |
|---|---|---|---|
| EMAC-L90 | -0.71 [-0.82; -0.54] | 0.29 [0.02; 0.52] | -0.39 [-0.59; -0.13] |
| EMAC-L47 | -0.60 [-0.82; -0.54] | 0.13 [-0.15; 0.39] | -0.53 [-0.38; -0.74] |
| EMAC-L47-o | -0.76 [-0.86; -0.61] | 0.34 [0.18; 0.56] | -0.63 [-0.77; -0.43] |
| WACCM -1 | -0.56 [-0.72; -0.34] | 0.22 [-0.05; 0.46] | -0.45 [-0.64; -0.20] |
| WACCM -2 | -0.61 [-0.75; -0.40] | 0.49 [0.15; 0.61] | -0.64 [-0.78; -0.45] |
| WACCM -3 | -0.60 [-0.75; -0.40] | 0.49 [0.16; 0.61] | -0.67 [-0.79; -0.48] |
| WACCM -4 | -0.85 [-0.91;-0.76] | 0.44 [0.11; 0.58] | -0.39 [-0.59; -0.13] |
| GEOSCCM | -0.59 [-0 74; -0.39] | 0.74 [0.59; 0.84] | -0.57 [-0.73; -0.35] |
| NIES | -0.78 [-0.87; -0.64] | 0.40 [0.153; 0.61] | -0.22 [-0.46; 0.06] |
| SOCOLv3 | -0.57 [-0.73; -0.351] | 0.33 [0.07; 0.55] | -0.59 [-0.74; -0.38] |
| MRI | -0.81 [-0.88; -0.69] | 0.43 [0.17; 0.63] | -0.29 [-0.52; -0.02] |
| ACCESS -2 | -0.54 [-0.71; -0.32] | 0.17 [-0.11; 0.42] | -0.50 [-0.68; -0.27] |
| ACCESS -1 | -0.58 [-0.74; -0.37] | 0.14 [-0.14; 0.39] | -0.51 [-0.68; -0.26] |
| CMAM | -0.67 [-0.8; -0.50] | 0.39 [0.14; 0.60] | -0.24 [-0.48; 0.03] |
| NIWA -1 | -0.30 [-0.53; -0.03] | -0.02 [-0.29; 0.25] | -0.62 [-0.76; -0.43] |
| NIWA -2 | -0.37 [-0.58; -0.11] | 0.33 [0.053; 0.54] | -0.70 [-0.82; -0.54] |
| NIWA -3 | -0.29 [-0.52; -0.02] | 0.11 [-0.17; 0.37] | -0.46 [-0.65; -0.21] |
| NIWA -4 | -0.33 [-0.55; -0.08] | 0.35 [0.09; 0.56] | -0.64 [-0.78; -0.45] |
| NIWA -5 | -0.11 [-0.36; 0.17] | 0.08 [-0.19; 0.35] | -0.37 [-0.58; -0.11] |
| ULAQ -1 | 0.34 [0.07; 0.56] | -0.25 [-0.49; 0.02] | -0.75 [-0.85; -0.61] |
| ULAQ -2 | 0.30 [0.03; 0.52] | -0.15 [-0.40; 0.13] | -0.67 [-0.80; -0.49] |
| ULAQ -3 | 0.38 [0.12;0.59] | -0.19 [-0.43; 0.08] | -0.80 [-0.88; -0.68] |
| MMM | **-0.56±0.19** | **0.31±0.18** | **-0.49±0.15** |
| observations | | | -0.21 [-0.45; 0.07] |

these upper regions). Overall, even for the short periods of about 20 years length for which we have observational data (i.e. period 1998-2016 and the slightly longer period 1998-2018), the ozone trends are consistently positive for the models and the observations, indicating that the upper stratosphere MMM trend is robust to inter-annual variability and we are likely observing the forced signal driven by GHG and ODS changes.

5    For the lower stratosphere, Fig. 8b again highlights that ozone trends are highly variable in particular for the shorter periods of about 20 years and that the MMM ozone trends over all periods are negative in the tropics and positive in the mid-latitudes




in the REF-C2 simulations. In general, the mid-latitude ozone trends are very variable both in the northern and southern mid-latitudes, but the southern mid-latitude trends are somewhat lower (and negative in some models) for the shorter periods. Also in observations, the SH mid-latitude trend is more uncertain and variable (compare observational estimates in Fig. 8b, and see Ball et al., 2019a). Fig. 8b further highlights again the dipole structure in LS ozone trends, i.e., negative ozone trends in the

tropics are often accompanied by northern and southern mid-latitude ozone trends of opposite sign. This dipole trend pattern in models disagrees with the robust observed ozone trend pattern that implies negative trends in both tropics and mid-latitudes. Note that the mid-latitude ozone trend of the fGHG simulations will discussed below.

As mentioned above, there are a number of possible explanations for the disagreement of the mid-latitude ozone trends between models and observations:

– If assuming that modeled trend distributions are correct, the observed ozone trend as an unlikely representation might emerge due to very anomalous conditions during the regarded periods, caused for example by extrema in natural variability in the beginning of the time series (late 1990s), and/or in the end of the time series (late 2010s).

– The modeled trend distribution constructed from the REF-C2 simulations might be biased because natural variability (e.g. QBO and ENSO) is not represented adequately in the models. This could lead to a too narrow trend distribution,

15        and thus would make the observed trend seem more unlikely than it is.

– The modeled trend distributions might be incorrect in their mean value, i.e., the forced trend might not be captured correctly by the models.

While it is not easily possible to test which of the above explanations is correct, we will discuss each of them in the following.

**Influence of natural variability on the observed trend**

Sources of natural variability that strongly influence LS ozone are volcanic eruptions, QBO and ENSO. No major volcanic eruption occurred during the analyzed period, so we will disregard this source of variability. The influence of QBO and ENSO on the hemispheric mean mid-latitude ozone is of the same magnitude, and thus they can both impact LS ozone trends, as shown by the study of Olsen et al. (2019).

We know from earlier studies that the QBO has a strong dynamical effect on the sub-tropical and mid-latitude LS ozone (e.g.

Randel and Wu, 2007). Moreover it was recently shown that ozone trends in the mid-latitudes are directly linked to the QBO, as the QBO induces a secondary circulation (see e.g. Ball et al. (2019a) and A. Stenke personal communication, EGU 2020). In 2016, the typical QBO phasing was disrupted, and this has been shown to be associated with negative LS ozone anomalies in the tropics (Kusuma et al., 2019). These negative anomalies at the end of the time period would lead to a strengthened negative ozone trends, and our analysis indeed shows slightly stronger negative tropical ozone trends for the end year 2016 compared

to 2015 (see Fig. 4a). The mid-latitude ozone trend is also stronger for the end year 2016, which however does not fit the expectations (QBO-induced anomalies are of different sign in tropics and extratropics, see e.g. Randel and Wu, 2007). Another way, in which the QBO could lead to decadal scale variability in ozonr, and thus influence the trends, was recently reported



(J. Neu personal communication, AGU 2018): since the QBO's influence on tropical upwelling depends on the season, the timing of the QBO phases is crucial for it's influence on trace gas concentrations. Similarly, Ball et al. (2019a) pointed out that non-linear attribution may be required to capture the QBO's impact.

One of the strongest warm ENSO events on record occurred in late 1997 (Jensen et al., 1998). By using CCM (WACCM) simulations with prescribed SSTs from observations, Calvo et al. (2010) showed that this strong ENSO event was associated with low ozone values in the tropics and high values in the mid-latitudes. This is in line with observational results by Randel et al. (2009). Consequently, mid-latitude ozone trends should be more negative when beginning the time period with this warm ENSO year. This is consistent with the strong mid-latitude trends in the BASIC$_{SG}$ data-set for the start years 1998 (and less so 1996-1997, see Fig.4 (b)). However, as the tropical trend is not associated with weaker negative trends for the start year 1998, this explanation is again not fully consistent.

As stated earlier, we have refrained form applying a multiple linear regression (MLR), which potentially would take the named sources of variability into account. If the trend strengths and patterns are strongly influenced by anomalous natural variability events, one might argue that removing this variability via an MLR method would have a large impact on the trends. However, the trend estimates by Ball et al. (2018), that take ENSO and QBO variability into account differ only in details to our trend estimates. Note that an MLR method might not fully account for the induced signals by QBO or ENSO, because, as mentioned above, their influence is likely non-linearly dependent on the signal strength and the signal timing. Thus, also an MLR analysis cannot conclusively clarify the role of natural variability to lead to the observed trends.

Overall, the sudden systematic change in the magnitude of the mid-latitude observational trend (Fig. 4 (b)), indicates that natural variability (in particular the strong ENSO event in 1997) influenced the observed trends over the analyzed periods, and contributed to the particular strong disagreement of observed and modeled mid-latitude trends for the relevant time periods. However, the expected effects of QBO and ENSO events on the trends are not entirely consistent between tropics and mid-latitudes. Possibly an exceptional combination of different factors led up to the particular observed trend pattern, causing the mid-latitude trends to be more anomalous than the tropical trends in comparison to the trend distribution derived from the models.

**Representation of natural variability in models**

Above, we argued that natural variability likely influenced the observed trends, and that might partly explain that trends over the observed period disagree with model simulations when compared to modeled trends. However, how large this disagreement is, depends on the underlying trend distribution derived from the models. E.g. if the influence of natural variability is underestimated in the models, the trend distribution is too narrow.

The QBO is represented differently in the individual CCMs: some models generate a QBO internally, some models nudge winds towards a given QBO, and in some models, the representation of the QBO is missing entirely (for more details see Morgenstern et al., 2017). Thus, over the whole suite of models, this could cause an underestimation of ozone variability in the models and therewith a too narrow trend distribution. Moreover, as the QBO signal is treated differently across the REF-C2





model setups, we can also expect that the inter-model differences in the QBO representation contributes to the spread in ozone trends over the post-ODS period.

The analyzed free running REF-C2 simulations either use an interactive ocean model, or use SSTs from other model simulations, that are coupled to an ocean model. However, these coupled models still have biases with respect to the simulation of
ENSO (Bellenger et al., 2014), thus ENSO-related variability in LS ozone might also be underrepresented.

Further, even if QBO and ENSO are represented with correct signal strength (e.g., by nudging the QBO, and prescribing observed SSTs), the induced circulation anomalies might not be captured entirely by the models. Thus, even if hindcast simulations with prescribed observed SSTs were used, this does not guarantee that the effects of natural variability on ozone trends are fully captured. It would be interesting to compare the modeled trend distributions from the REF-C2 simulations to such
hindcast simulations (REF-C1), however, the data of those hindcast simulations is only available until 2010.

**Representation of the forced trends in models**

One of the reasons for comparing the observed and modeled ozone trends, is to infer whether our current understanding on ozone trends forced by changes in GHG and ODS concentrations is correct, or whether state-of-the-art models are missing relevant processes.

The contribution of GHG and ODS changes to the LS ozone trends can be inferred from the fGHG simulations, as shown in Fig. 8b. We find that the MMM mid-latitude ozone trends are nearly similar between the fGHG and the REF-C2 simulations for short time periods of about 20 years. For the LS tropics the trends are weaker in fGHG simulations, as expected from the missing influence of GHG increases on tropical upwelling, but they are not significantly different from each other. The small differences between fGHG and REF-C2 trends over the observational period again underlines the conclusion that variability
strongly impacts LS ozone trends.

For the longer time periods, the MMM fGHG trend in the tropical LS is slightly positive, and can be clearly distinguished from the REF-C2 trend. This can be explained by the absence of the GHG induced enhancement of tropical upwelling, which strongly influences tropical LS ozone trends. In the mid-latitudes, the MMM trend of the fGHG simulations is somewhat smaller, but not significantly different from the REF-C2 trend, indicating that enhanced downwelling associated with the
strengthened circulation plays a minor role. This weak influence of downwelling trends on mid-latitude ozone trends for the particular chosen periods here are consistent with the results presented in Section 3.4, where we found a small positive, but insignificant and non-robust relationship between tropical upwelling (used as proxy for extratropical downwelling) and mid-latitude ozone trends in REF-C2 (see Tab. 2). These weak correlations indicate that other processes, e.g. two-way mixing, influence the mid-latitude model trends.

The studies of Wargan et al. (2018) and Ball et al. (2019b) argue that the LS mid-latitude ozone decrease in observational data is possibly linked to enhanced two way mixing. Ball et al. (2019b) used effective diffusivity (Haynes and Shuckburgh, 2000) as diagnostic for horizontal mixing and found that in reanalysis data (JRA-55, ERA-Interim) mixing is enhanced in the post-ODS period. Ray et al. (2010) also showed in an earlier study a substantial increase in effective diffusivity under





a changing climate for CCMs and reanalyses data (JRA-25, ERA-40). Recently, Orbe et al. (2020) used the TEM budget analysis of an idealized short lived tracer (that covaries with ozone on interannual and decadal time scales) in 10 free running ensemble member simulations with the GEOSCCM model, in order to identify the mechanism that is driving the negative LS ozone trends. It is very interesting to note that they found negative LS ozone trends in the northern sub-tropics (20°-40°N, 70-100hPa) and in the mid-latitudes (40°-60°N, 100-200hPa) in the GEOSCCM simulation for the winter seasonal mean, however the trends are weak compared to observations. In our analysis the GEOSCCM simulation shows a weak positive LS trend in the tropics and a sligthly negative trend in the mid-latitudes (see Fig. 1). However we cannot compare our trends directly to those of Orbe et al. (2020), as we use different latitude boxes, calculate the trend over all seasons, and do not use the same free running simulations. In contrast to the studies of Ball et al. (2019a) and Wargan et al. (2018), the study by Orbe et al. (2020) showed that the mixing effect is not as important for the LS mid-latitude ozone trend. Rather they found a poleward expansion of the residual circulation in the LS with weaker downwelling in the subtropics, and stronger downwelling over the mid-latitudes, leading to negative LS trends in the NH. However, as discussed in Orbe et al. (2020), mixing must be considered in the context of the specific tracer that is analyzed (i.e. short lived travers are less sensitive to mixing). As such, the analysis of the TEM budget for the tracer ozone could be a focus in further investigations.

Whether the processes such as mixing, or shifts in the downwelling region are reflected in the analyzed trends might be a matter of the choice of the averaging regions. In our study, we selected the tropical box as ranging from 20°N-20°S, 30-100 hPa and the NH mid-latitude box as ranging from 30°-50°N and 30-150 hPa, but those exact choices are rather arbitrary. Trends and inter-model and interannual correlation coefficients are sensitive to the choice of these boxes. For LS ozone trends it is obvious from Fig. 1, that varying the boxes can lead to different trends. This should especially be the case for the the correlation between tropical upwelling and mid-latitude ozone, as the region of downwelling varies for different models (Chrysanthou et al., 2019). We also found that the mid-latitude correlation varies with the slightly different choice of post-ODS period. One reason for this lack of robustness in the correlation coefficients at all levels (see profiles in Fig. 6b) could be again the fact that the region of downwelling is variable, and therefore, the latitudes where LS ozone is affected by this process changes over time and differs from model to model. This means that the selected box is probably better for the correlation over a certain periods. As mentioned above, the study of Orbe et al. (2020) found an expansion of the region of mean downwelling during the post-ODS period for GEOSCCM ensemble members and this supports our findings that different periods can have different correlations. However another reason for the less robust correlations could be the models' different influence of the QBO phase on the LS ozone trends, or the combined effects of GHG-induced circulation strengthening and of recovery from ODS (see discussions above).

Overall, the LS ozone trends are strongly affected by variability, making it difficult to infer whether the forced trends in models and observations agree. For the models, we extended the time period into the future to investigate for which period length the trends converge. We find that the inter-model spread of the ozone trends substantially diminishes for longer time periods (i.e., 1998–2030 and 1998–2040), but to a different extent for different regions (see Fig. 8). In the upper stratosphere, MMM trends are significantly positive with all end point years. In the LS, the MMM ozone trends consistently show a persistent dipole trend





pattern over all periods. However, as the model spread in the trends is especially large in the mid-latitudes over the short time periods, the dipol trend pattern is not significant there. As expected, the model spread in the trends is reduced when extending the time period of analysis. For the trend estimate over the years 1998–2040 we have a significant dipole trend pattern with a comparably low inter-model spread. From the individual REF-C2 simulations (Fig. 5) we learned that all model simulations

evolve the dipole trend pattern over long time periods. Thus the question arises as to whether we can expect that observational data also evolve the dipole trend pattern in the future. If the forced model trends are correct, we should expect the dipole pattern to emerge by about 2030 to 2040 (compare Fig. 5). However, to judge whether the forced ozone trends disagree between models and observations before that, it will be essential to gain a better understanding of the different processes driving variability and trends in mid-latitude LS ozone.

## 5  Conclusions

In the present study, we analyzed in detail trends in stratospheric ozone for the recent post-ODS period 1998–2018, and for slightly different chosen post-ODS periods, using a total of 31 simulations of different state-of-the-art chemistry climate models and compared them to the observational based dataset $BASIC_{SG}$. Moreover, we linked the ozone trends to tropical upwelling and discussed the reasons for the differences in the LS ozone trends between models and observations. The main findings of

our study are summarized in the following.

1) LS ozone trends over the period 1998–2018 vary strongly across different models and among different ensemble members of the same model setup. Thus internal variability strongly influences the LS ozone trends over this short time period. But even if this high variability is taken into account, none of the model simulations reproduces the overall negative observational trend pattern which extends from SH to NH mid-latitudes. Thus the observed LS ozone trend pattern is a rather unlikely realization

in state-of-the-art CCM simulations.

2) The models' LS ozone trend (given as the most likely values of the models' trend probability distribution) remains negative in the tropics and positive in the mid-latitudes even when varying the time period between 1995 and 2019. Although there is quite a large spread in the trend magnitude, these modeled trends do not show a systematic change within the different periods. For observations, LS trends remain negative in both the tropics and the mid-latitudes. In contrast to the models' consistent

trend we find a systematic shift in the trend magnitude towards less negative mid-latitude trends for start years 1999 and 2000, although trends remain negative for all start/end year combinations.

3) We further find that in the tropics the observed trends are likely represented by the models' trend distribution. However in the mid-latitudes the observational trends are a rather extreme value of the models' distribution.

4) All models evolve a dipole trend pattern, with negative trends in the tropics and positive trends in the mid-latitudes, for

longer time periods, but the models differ in their timing of convergence. As internal ozone variability behaves in a similar way in models and observations (indicating that processes driving variability are also similar), we probably can expect a dipole trend pattern for observations future, too.

5) The influence of tropical upwelling on the tropical LS ozone trend is high, this is in line with many studies before. The mid-latitude LS ozone trend is less highly linked to tropical upwelling/downwelling, indicating that other processes, e.g. two

way mixing, play a role.





Finally we discussed the question whether the apparent discrepancy between model and observational trends is due to the misrepresentation of certain processes in the models (e.g. the mixing strength, residual circulation strength) or due to not adequately represented natural variability (ENSO/QBO) in models, or whether the observational trend is just an extreme (but plausible) realization of the models' distribution. If model processes or variability are inadequately represented, the next question is, if they could cause uniform negative ozone trends in the LS. Thus to get a better quantitative understanding of the processes that cause LS ozone trends in climate models it is important to explicitly analyze the transport processes that are dominating the LS ozone trend.

*Author contributions.* SD performed the data analysis and produced the figures. SD, RE and HG made substantial contributions to conception and design, analysis and interpretation of the data. WTB provided the observational ozone trends and contributed to the interpretation of the results. Moreover all authors participated in drafting the article.

DATA: All data of CCMVal-2 and CCMI-1 used in this study can be obtained through the British Atmospheric Data Centre (BADC) archive (ftp://ftp.ceda.ac.uk). CESM1-WACCM data have been downloaded from http://www.earthsystemgrid.org. For instructions for access to both archives see http://blogs.reading.ac. uk/ccmi/badc-data-access

The authors declare that they have no conflict of interest.

*Acknowledgements.* This study was funded by the Helmholtz Association under grant VH-NG-1014 (Helmholtz-Hochschul- Nachwuchs-forschergruppe MACCClim). We acknowledge the modeling groups for making their simulations available for this analysis, the joint WCRP SPARC/IGAC Chemistry-Climate Model Initiative (CCMI) for organizing and coordinating the model data analysis activity, and the British Atmospheric Data Centre (BADC) for collecting and archiving the CCMI model output. We thank Marta Abalos for providing us an additional (forth) ensemble member of WACCM. Moreover, we want to acknowledge that the EMAC simulations were done within the project ESCiMo (Earth System Chemistry integrated Modelling), a national (German) contribution to the ChemistryClimate Model Initiative, and have been performed at the German Climate Computing Centre DKRZ through support from the Bundesministerium fuer Bildung und Forschung (BMBF). W.T.B. was funded by the SNSF project 200020_182239 (POLE)". BASIC$_{SG}$ for 1985-2019 will be available for download from https://data.mendeley.com/datasets/2mgx2xzzpk/4 following review of this paper. GOZCARDS ozone data contributions from L. Froidevaux, R. Wang, J. Anderson, and R. A. Fuller at the Jet Propulsion Laboratory are gratefully acknowledged.



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





**Figure 4.** Tropical (right) and mid-latitude (left) LS ozone trends (in DU/dec) as function of different periods for the observational trend of BASIC$_{SD}$ ((a)+(b)), the most likely trend of the modeled REF-C2 probability distribution ((c)+(d)) and the 1-$\sigma$ standard deviation (in DU/dec) of the mean obtained from the probability distribution ((e)+(f)). The panels (g)+(h) show the "probability of disagreement" (in %) between observed trends and the REF-C2 trend probability distribution. In all panels the x-coordinate denotes the different end years (2013-2019) and the y-coordinate the different start years (1995-2001).







**Figure 5.** Tropical (20°S-20°N) and northern mid-latitude (30°N-50°N) LS ozone column trend (in DU/dec) of observations and of REF-C2 model simulations as a function of the end year (red and blue dots, respectively). Tropical upwelling trend is included for all REF-C2 simulations, where TEM diagnostics was available (black dots). The end year varies from 2013 to 2019 for observational data and from 2013 to 2060 for REF-C2 simulations. Error bars associated with each trend represent the 90% confidence intervals.



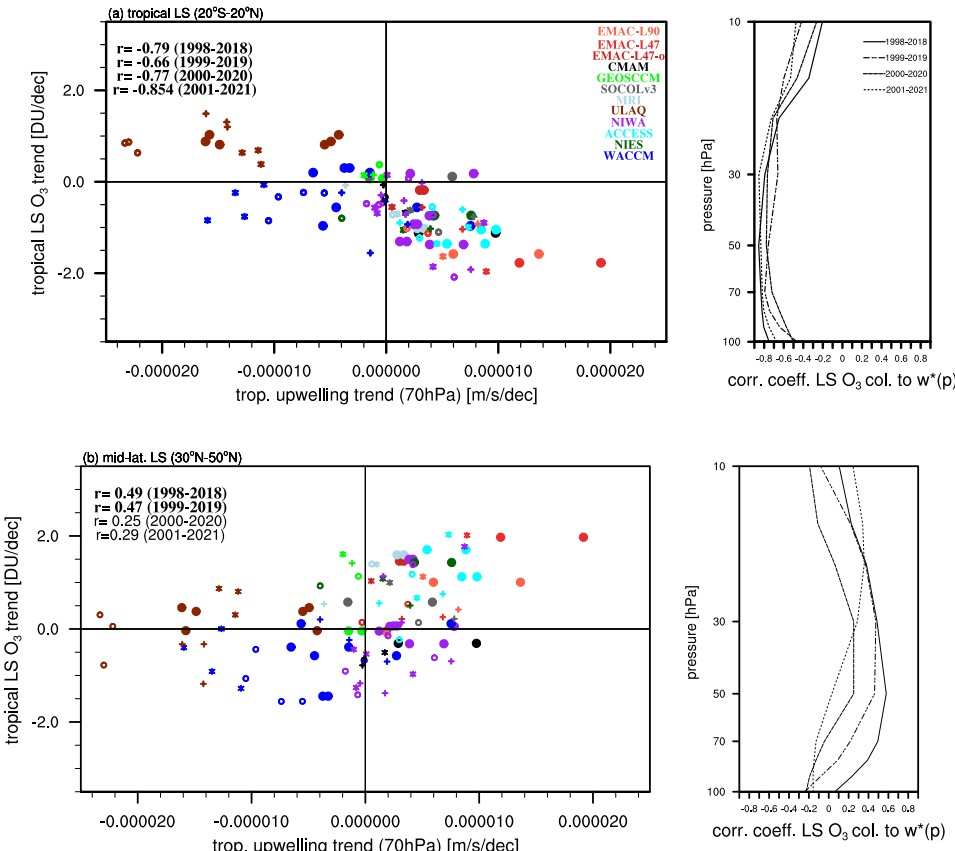

**Figure 6.** Intermodel correlation over 22 simulations (a) between tropical upwelling trend at 70 hPa and tropical (20°S-20°N) LS ozone column trend and (b) between tropical upwelling trend at 30 hPa and northern mid-latitude (30°N-50°N) LS ozone column trend. Trends are calculated over the four different post-ODS periods 1998-2018, 1999-2019, 2000-2020 and 2001-2021. In the left panels LS ozone trends (in DU/dec) are plotted against tropical upwelling trends. The same color represents one CCMI model (thus ensemble simulations are given in the same color). The corresponding correlation coefficient (for the years 1998-2018, 1999-2019, 2000-2020, 2001-2021) are given within the panels and are highlighted in bold, if they are significant at the 95%-level. In the right panels the inter-model correlation coefficients of LS ozone trend and tropical upwelling trends at 10, 30, 50, 70, 100 hPa are shown.







**Figure 7.** Interannual correlation coefficients between the tropical LS ozone column (20°N-20°S, 30-100 hPa) and the local zonal mean ozone values in observations and in REF-C2 simulations. Correlation coefficients are based on detrended yearly mean ozone data over the time period 1998-2060 for CCM data and over the time period 1998–2018 for observational data. Boxes illustrate LS regions in the tropics and the northern mid-latitudes.


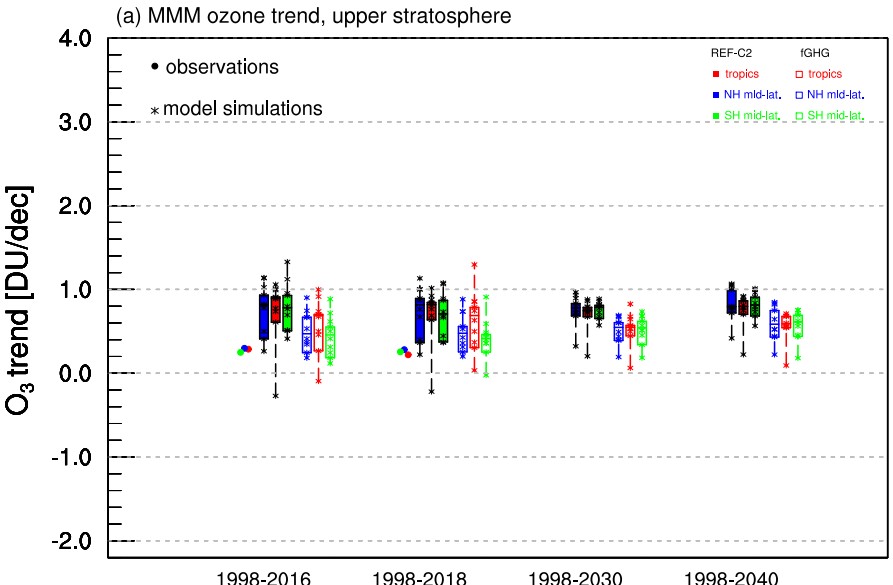

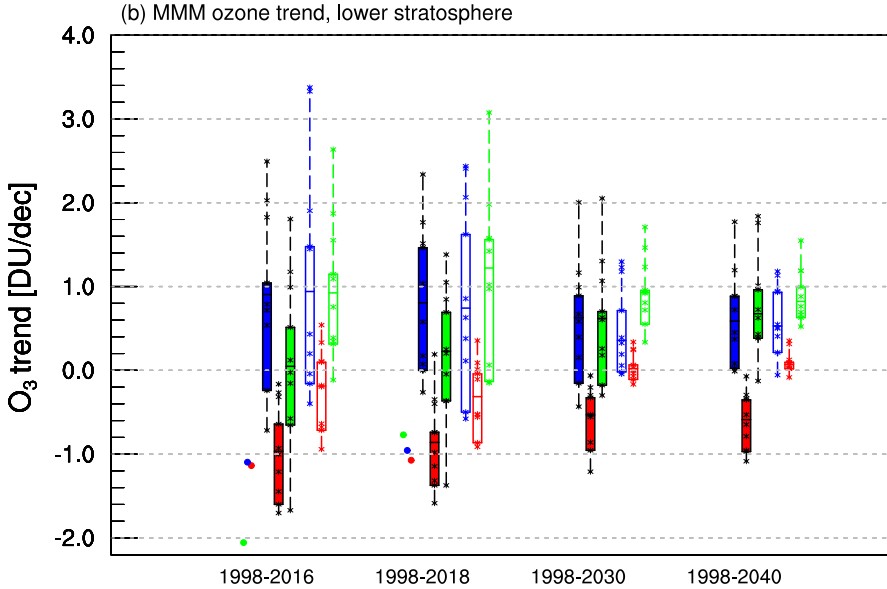

**Figure 8.** Multi-model-mean ozone column trend in the tropics (red, 20°N-20°S), in the northern mid-latitudes (blue, 30°-50°N) and in the southern mid-latitudes (green, 30°-50°S) as a function of time for four different post-ODS periods (1998-2016, 1998-2018 1998-2030 1998-2040) for (a) the upper stratosphere (1-10 hPa) and (b) the lower stratosphere (30-100/150 hPa). The boxes extend from the lower to upper quartile of the data with a line for the median and with whiskers to show the minimum and maximum values of the LS MMM ozone trends. Moreover the individual model trends are denoted by black stars. MMM trend estimates are given for REF-C2 simulations (filled boxes) as well as for fixed GHG simulations (not-filled boxes). Note here that for the estimate of the MMM trends only 10 model simulations are taken into account, as the fixed GHG simulations have this smaller sample size, and we want to ensure that both simulations types (fGHG and REF-C2) include the same models for the MMM trend estimate. Moreover we exclude the outlier model ULAQ for the MMM. Observational data are included for the trends over the period 1998-2016 and 1998-2018 (red, blue, green points, respectively).