# Peer review of "Analysis of recent lower stratospheric ozone trends in chemistry climate models"

_Atmospheric Chemistry and Physics, 2020_

## Referee Comment (RC1) · Anonymous Referee #1 · 13 Nov 2020

In this study the authors build on previous studies' examination of recent (1998-2018) trends in stratospheric ozone, in particular those spanning the middle and lower stratosphere. The main contribution from this study, which distinguishes it from previous work, is to put the results from the previous single-model and observational studies in the broader context of (several) more models. Through analysis of the ozone trends in a 31-member multi-model ensemble of free-running simulations of the recent past the authors quantify the model uncertainty in ozone trends, placing the observations in that context via construction of a "probability of disagreement" distribution, and test various sensitivities to these trends with respect to the end years of the time series and location in the stratosphere. An analysis of the convergence of the trends into the future is also made as well as a basic analysis ascertaining the extent to which modeled trends in

ozone are related to trends in upwelling. The work is very detailed and rigorous with respect to testing sensitivities, etc. The focus on the latter makes sense in the context of previous studies and given the short time series of the period of focus. At the same time, however, I have same major comments that explain why I have assigned to this study "major revisions" which are as follows:

1. The discussion of a forced signal (driven by GHG vs. ODS changes) presented in Section 4 is somewhat lacking, especially given the facility with which the authors can bring in the results not just of the fGHG simulations that they have analyzed but also the fODS simulations that were also performed for the REF-C2 scenario. Both, for example, were employed in the study by Abalos et al. (2019) cited below. By explicitly looking at these simulations, in addition to the two others considered here, the authors can more quantitatively address the relative roles of ODS vs. GHG (and the linearity between their interactions). This is a reasonable request especially given that ODS themselves can alter the stratospheric circulation (see the impacts on upwelling documented in the second study listed below) and given that these experiments have already been performed.

Abalos, Marta, Clara Orbe, Douglas E. Kinnison, David Plummer, Luke D. Oman, Patrick Jöckel, Olaf Morgenstern et al. "Future trends in stratosphere-to-troposphere transport in CCMI models." Atmospheric Chemistry and Physics 20, no. 11 (2020): 6883-6901.

Abalos, Marta, Lorenzo Polvani, Natalia Calvo, Douglas Kinnison, Felix Ploeger, William Randel, and Susan Solomon. "New Insights on the Impact of Ozone‐Depleting Substances on the Brewer‐Dobson Circulation." Journal of Geophysical Research: Atmospheres 124, no. 5 (2019): 2435-2451.

2. The discussion of the mechanism underlying the different ozone trends is a bit unsatisfying. Of course, most of this derives from focusing on a multi-model comparison for which it is (understandably) difficult to do a detailed budget analysis for each model.

However, the authors have more information than they may realize. In particular, I would strongly encourage the authors to consider analyzing the "age of air" or "e90" tracers that were also carried in these integrations as these provide a description of the actual transport circulation changes simulated in the models (which may, or may not, be directly related to changes in upwelling). The lack of any passive tracer diagnostic is a bit discouraging and I think it's incorporation would add substantially to the discussion.

3. In contrast to the previous sections, I find much of the material in Section 4 to be qualitative and speculative. For example, it is, of course, true that intermodal differences in internal variability (contributed from the QBO and ENSO) can contribute to the spread in trends among the models. However, this is never explicitly shown (only described in generalities) and I think a basic analysis needs to be done by which, for example, the authors select two models with very different ozone trends over midlatitudes and then show their ozone composites with respect to different phases of ENSO and the QBO. How does the ozone variance contributed from these two modes vary across models? Is it large? This would be an easy calculation to do and could be provided as a supplementary figure. Without a more quantitative analysis, though, it is not clear what exactly is gained from this discussion, besides raising issues that have been discussed in previous studies.

4. It appears that one of the main results from this study is, per the conclusions, the fact that "in midlatitudes the observational trends are a rather extreme value of the models' distribution." I agree with the authors that this is an important conclusion and I think this is a nice finding from this study. However, I think the authors need to acknowledge that this was also the conclusion made in Orbe et al. (2020) (https://agupubs.onlinelibrary.wiley.com/doi/full/10.1029/2019JD031631). Specifically, please find in the abstract the statement "while the free‐running model can produce negative ozone changes in the NH LS, the magnitude of these changes is significantly weaker, relative to both M2GMI and MERRA‐2; moreover, these weaker ozone

decreases are consistent with weaker simulated changes in the residual circulation." That study, therefore, arrived at a very similar conclusion to the one obtained in this study, albeit using an ensemble of (single model) simulations and not a multi-model ensemble. I strongly encourage the authors, therefore, to better cite this contribution earlier in the manuscript. In particular, on page 2 in their reference to this study they can append an additional statement along the lines of "…is primarily associated with large-scale advection; furthermore, they showed that the observed changes in advection and ozone are within the range of model variability (gauged from one CCM) but on the extreme end." The conclusion made in Orbe et al. (2020), therefore, is distinct from that in Chipperfield et al. (2018) who suggested that the observed ozone trends were well within the range of simulated variability.

5. Quite a bit of attention is paid to the correlation between tropical ozone trends and midlatitude ozone trends. This is understandable, given that the two are plausibly connected, but Figure 3 does not really seem to support this. The correlation seems very small, no? I think the reader would find this relationship more convincing if the authors showed a figure showing this relationship for, say, a given model. In particular, does this relationship manifest by just considering interannual variability? What does the correlation between midlatitude and tropical ozone look like for individual years within a given model? Without a stronger case it just seems like Figure 3 is exhibiting a very weak relationship. . ...

6. Page 18, Lines 7-26: A lot of ambiguity and potential for intermodel differences is described here as stemming from differences in the latitudinal extent of upwelling/downwelling between models. I certainly agree with this comment. However, there is a very straightforward solution. One could compare w* between models in such a way that accounts for intermodel differences in the turnaround latitudes of the BDC. In particular, it is possible that the fixed lattiude boxes considered here do not span the region of mean downwelling in every model owing to differences in the meridional extent of the BDC. Not accounting for this information, therefore, would lead to the misleading conclusion that the models somehow underestimate downwelling but, actually, this may not be the case since the model may simply have downwelling occurring at different latitudes. What happens when you redo your analysis to be more dynamically consistent in this regard?

In addition to the major comments above I also provided these more minor points:

-Page 6, Line 20: Are different ensemble members treated the same/given the same weight as different models? Shouldn't they be weighted in such a way that distinguishes between ensemble members versus distinct models? Perhaps that is what has been done — it is not clear in the present text, however. -Page 7, Line 13: "dynamical linear modeling" needs to be described here. -Page 8, Line 13: The Orbe et al. (2020) study also showed this discrepancy in the LS ozone trend between the observations and the models. -Figure 5: Can you add the observed trends in upwelling as well? This seems important. Of course there may be differences between reanalyses but you can add, for example, estimates from MERRA-2 and ERA-Interim. This should be easy to do as you can use the TEM residual circulation estimates from the SPARC Reanalysis Intercomparison Project (https://s-rip.ees.hokudai.ac.jp/resources/data.html). -Page 24, Line 6: It is not clear to me what the discrepancy is here that you are claiming between the GEOSCCM results presented in that study compared to the ones the authors show in Figure 1. Please explain in more detail. -The language throughout could be improved at various places. I have noted a few grammatical errors below but there are many others. I strongly encourage the main author to have all co-authors check for lingering language issues/typos.

Technical Points:

-Page 1, Line 7: Please indicate a reference for CCMI -Page 2, Line 6: "results from" -> "result from" -Various paragraphs throughout are not indented which renders the formatting a bit awkward (e.g. Page 12, Line 5). Please fix. -Page 12, Line 3: "depending on" -> "dependent on" -Page 12, Line 11: Do you need to remove "not" in front of

significant? This is confusing given that the next sentence implies that the trends are significantly related. -Page 15, Line 3: The sentence starting with "We will show. . ." is not complete. -Page 15, Line 15: "evolve" -> do you mean "simulate"?

---

## Referee Comment (RC2) · Anonymous Referee #2 · 17 Nov 2020

This paper addresses the recently reported (Ball et al. 2018) widespread decline in lower stratospheric ozone at low and mid-latitudes. This is a high profile topic in ozone-layer science. The paper presents a comprehensive analysis of state-of-the-art CCM simulations to investigate if, and to what extent, different simulations capture the observed trends. The models show some agreement in the tropics but generally fail to capture the combined tropical and mid-latitude trends. Although the paper does not resolve the cause of the observed trends, it does show the ability of current CCMs and point the way to further studies. Overall I think that the paper is a very useful contribution to this topic and well suited for ACP. I recommend publishing after addressing the minor comments below.

1) There are a lot of minor editorial issues, especially missing hyphens (e.g. 'freefrunning') and missing commas. I assume that the editorial office will sort these out if the authors don't find them.

2) Page 1. Line 10-11. It is not clear what the 'systematic change' relates to. The sentence mentions 'different analysis periods', but for the change to be systematic the period would have to be changing in a particular direction?

3) Page 2. Line 4. Need to say '1987 Montreal Protocol and later Adjustments/Amendments'. Also, different ODSs started to decline at different times. Some (e.g. HCFCs) might still be growing. You mean the total halogen loading from ODSs.

4) Page 2. Line 8. There were still ODSs in atmosphere in 1980. Need to say 'pre-1980' or similar.

5) Page 3. Line 12. Put references in chronological order.

6) Page 4. Line 14. Ball et al (2018) also included some SD runs which showed very poor agreement with the observed trends. That could be mentioned here.

7) Page 5. Footnote. 'fourth'.

8) Page 8. Line 22. Please state which model was used by Stone et al. Is that one of the CCMI models?

9) Page 9. Caption (and elsewhere). Post 1998 is not the 'post ODS' period. ODSs are still present and different ones have different trends. Total chlorine and bromine are declining, which is not the same thing. You should find another description.

10) Page 12. Line 3. 'depend'.

11) Page 12. Line 11. 'low not significant'. This reads strangely. Maybe it is a lack of a comma, but could also be better to say 'non-significant'.

---

## Author Comment (AC2) · 17 Feb 2021

We thank referee #2 for the positive and constructive comments on our manuscript. Due to the suggestions of referee #1 we include some additional analysis to this paper. With that we gained additional insight to the possible processes determining the ozone trends in the LS. Due to the new results we also reorganized the structure of the manuscript in the last sections (see new section 3.4 and 3.5). Moreover, note that we do not consider interannual correlations any more, as we felt that not too much was learned here.

In the revised manuscript we consider all questions and comments.

Minor issues:

[Figure]

1) There are a lot of minor editorial issues, especially missing hyphens (e.g. 'free running') and miss-ing commas. I assume that the editorial office will sort these out if the authors don't find them.

-> We re-read the manuscript carefully and found some of these minor issues. If there are re-maining ones, they will surely be corrected during the typesetting process.

2) Page 1. Line 10-11. It is not clear what the 'systematic change' relates to. The sentence mentions 'different analysis periods', but for the change to be systematic the period would have to be chang-ing in a particular direction?

-> Rephrased.

3) Page 2. Line 4. Need to say '1987 Montreal Protocol and later Adjust-ments/Amendments'. Also, different ODSs started to decline at different times. Some (e.g. HCFCs) might still be growing. You mean the total halogen loading from ODSs.

-> Done.

4) Page 2. Line 8. There were still ODSs in atmosphere in 1980. Need to say 'pre-1980' or similar.

-> Done.

5) Page 3. Line 12. Put references in chronological order.

-> Done.

6) Page 4. Line 14. Ball et al (2018) also included some SD runs which showed very poor agreement with the observed trends. That could be mentioned here.

-> Done.

7) Page 5. Footnote. 'fourth'.

-> Done.

8) Page 8. Line 22. Please state which model was used by Stone et al. Is that one of the CCMI mod-els?

-> Done.

9) Page 9. Caption (and elsewhere). Post 1998 is not the 'post ODS' period. ODSs are still present and different ones have different trends. Total chlorine and bromine are declining, which is not the same thing. You should find another description.

-> Thank you for this helpful comment, we now do not use the expression post-ODS any more, which was indeed a min-nomer.

10) Page 12. Line 3. 'depend'.

-> Done.

11) Page 12. Line 11. 'low not significant'. This reads strangely. Maybe it is a lack of a comma, but could also be better to say 'non-significant'.

-> Done.

---

## Author Response (AR1)

**Reply to Anonymous Referee #1** (ACP-2020-947)

We thank referee #1 for the constructive comments on our manuscript and for his/her new ideas to improve this work. According to the referee's suggestions we now include some additional analysis to this paper (for details see below). Through these, we gained additional insight into the processes determining the ozone trends in the LS. Due to the new results we also reorganized the structure of the manuscript in the last sections (see new section 3.4 and 3.5). Moreover, note that we do not consider interannual correlations anymore, as we felt that not too much was learned here.

Below this reply, you find the revised manuscript considering all questions and comments. Additionally, we highlighted the changes of the manuscript for the pages 1-19, and attached them to the reply. From Section 3.4 on, the changes were so comprehensive, that it doesn't make sense to highlight them.

Major issues:

1.The discussion of a forced signal (driven by GHG vs. ODS changes) presented
in Section 4 is somewhat lacking, especially given the facility with which the authors
can bring in the results not just of the fGHG simulations that they have analyzed but
also the fODS simulations that were also performed for the REF-C2 scenario. Both,
for example, were employed in the study by Abalos et al. (2019) cited below. By
explicitly looking at these simulations, in addition to the two others considered here, the
authors can more quantitatively address the relative roles of ODS vs. GHG (and the
linearity between their interactions). This is a reasonable request especially given that
ODS themselves can alter the stratospheric circulation (see the impacts on upwelling
documented in the second study listed below) and given that these experiments have
already been performed

**-> Thank you for this helpful idea. We now included the fODS simulations to Figure 8a and 8b. Note that the ozone MMM trend has slightly changed compared to the last version of Fig. 8, as we now only took the simulations into account that provide ozone for both the fGHG and as well as the fODS simulations. Moreover, we changed the structure of the text, and included a new Section (Sec. 3.5) on the forced trends and their attribution (which was previously a part of the discussion).**

2. The discussion of the mechanism underlying the different ozone trends is a bit unsatisfying. Of
course, most of this derives from focusing on a multi-model comparison
for which it is (understandably) difficult to do a detailed budget analysis for each model.
However, the authors have more information than they may realize. In particular, I
would strongly encourage the authors to consider analyzing the "age of air" or "e90"
tracers that were also carried in these integrations as these provide a description of
the actual transport circulation changes simulated in the models (which may, or may
not, be directly related to changes in upwelling). The lack of any passive tracer diagnostic is a bit
discouraging and I think it's incorporation would add substantially to the discussion.

**-> Thank you for this excellent suggestion. Following your comment, we now included analysis of the intermodel correlations between local ozone trends to AoA trends, and their correlation is**

**indeed very strong in the tropical to mid-latitude lower stratosphere (new Fig. 7a). Besides that, in order to distangle the different transport mechanism, we also include the correlation to residual circulation transit time (RCTT) and Aging by mixing trends (see Fig. 7b+c). The revised Section 3.4 describes the results and the additional insights from those analysis.**

3. In contrast to the previous sections, I find much of the material in Section 4 to be
qualitative and speculative. For example, it is, of course, true that intermodal differences in internal
variability (contributed from the QBO and ENSO) can contribute to
the spread in trends among the models. However, this is never explicitly shown (only
described in generalities) and I think a basic analysis needs to be done by which, for
example, the authors select two models with very different ozone trends over midlatitudes and then
show their ozone composites with respect to different phases of ENSO
and the QBO. How does the ozone variance contributed from these two modes vary
across models? Is it large? This would be an easy calculation to do and could be
provided as a supplementary figure. Without a more quantitative analysis, though, it is
not clear what exactly is gained from this discussion, besides raising issues that have
been discussed in previous studies.

**-> The reviewer is right, in that the discussion here is only qualitative. However, it is not the scope of our study to investigate the role of natural variability (e.g. QBO or ENSO) for the spread in the LS ozone trends in detail – we only want to discuss our results in the light of what is known from literature.   Therefore, this is part of the discussion section. To make the discussion character of section 4 clearer, we included to the text the following sentence: '' While it is not easily possible to test which of the above explanations is correct, in the following we will discuss their possible contributions to the diagnosed disagreement in the light of our results and of what is known from literature.'' (see p. 29, line 8).  Moreover, we also added in the discussion paragraph "Representation of natural variability in models", that we leave the assessment of the representation of natural variability and its effects on ozone to future studies (p.32, line1).**

4. It appears that one of the main results from this study is, per the conclusions,
the fact that "in midlatitudes the observational trends are a rather extreme value of
the models' distribution." I agree with the authors that this is an important conclusion and I think this
is a nice finding from this study. However, I think the authors
need to acknowledge that this was also the conclusion made in Orbe et al. (2020). …..

**-> Agreed. The Orbe et. al 2020 paper was published just before we submitted our draft, thus we missed to include it at several instances. It is now cited at several places.**

5. Quite a bit of attention is paid to the correlation between tropical ozone trends
and midlatitude ozone trends. This is understandable, given that the two are plausibly
connected, but Figure 3 does not really seem to support this. The correlation seems
very small, no? I think the reader would find this relationship more convincing if the
authors showed a figure showing this relationship for, say, a given model. In particular,
does this relationship manifest by just considering interannual variability? What does
the correlation between midlatitude and tropical ozone look like for individual years

within a given model? Without a stronger case it just seems like Figure 3 is exhibiting a very weak relationship.

**-> We agree in that the relationship between tropical and mid-latitude ozone trends is weak. Indeed, as the reviewer stated, there was a certain expectation to find a relation (which is backed up by the inter-annual correlations we showed in the previous version of the manuscript, see old Section 3.5 and in particular old Table 3 that did provide the correlation of inter-annual variability for individual models). Given our new insights into the role of different transport pathways, and the insights from the single-forcing experiments (see new Sec. 3.4 und 3.5), we realized that the expectation of anti-correlation between tropical and extratropical ozone might be misleading. Therefore, we strongly de-emphasized this point throughout the manuscript, including also the removal of old Section 3.5. Instead we focus on the trends and their correlation to transport measures (see above, new Sec. 3.4). Nevertheless, we decided to keep Fig. 3 as it nicely illustrates the mutual distribution of tropical and mid-latitude trends, but revised the text accordingly (see p. 14, lines 6-11).**

6. Page 18, Lines 7-26: A lot of ambiguity and potential for intermodel differences is described here as stemming from differences in the latitudinal extent of upwelling/downwelling between models. I certainly agree with this comment. However, there is a very straightforward solution. One could compare w* between models in such a way that accounts for intermodel differences in the turnaround latitudes of the BDC. In particular, it is possible that the fixed lattiude boxes considered here do not span the region of mean downwelling in every model owing to differences in the meridional extent of the BDC. Not accounting for this information, therefore, would lead to the misleading conclusion that the models somehow underestimate downwelling but, actually, this may not be the case since the model may simply have downwelling occurring at different latitudes. What happens when you redo your analysis to be more dynamically consistent in this regard?

**->Thank you, that's a good point – we were aware of that problem and therefore decided to follow the reviewer's suggestion to define a dynamically consistent mid-latitude box by averaging the LS ozone column from the turnaround latitudes of the BDC to 50°N . We re-calculate the LS ozone trend for this box (see Tab. 2) and moreover we indeed find a stronger inter-model correlation of LS mid-latitude ozone trends to up- and downwelling trends for this dynamical box (for details see Section 3.4 and Fig. 6).**

Minor Points:

-Page 6, Line 20: Are different ensemble members treated the same/given the same weight as different models? Shouldn't they be weighted in such a way that distinguishes between ensemble members versus distinct models? Perhaps that is what has been done but it is not clear in the present text, however.

**-> See p.10, line 30: "Note that for the calculation of the MMM trend, we chose to weight all 31 simulations equally (i.e., not considering that some models have multiple ensemble members) because "the trend variations among ensemble members are as large as among the different models over this period ". However later on in Section 3.4 the MMM is now calculated as average**

**of ensemble-means from each model, as for the longer time-periods the forced trends outweight variability, so that the above argument does not hold anymore.**

-Page 7, Line 13: "dynamical linear modeling" needs to be described here.

**-> We added a brief description of the "dynamical linear modeling" now – see p.9 line 1-4.**

-Page 8, Line 13: The Orbe et al. (2020) study also showed this discrepancy in the LS ozone trend between the observations and the models.

**-> We include Orbe et al. 2020 to the citations.**

-Figure 5: Can you add the observed trends in upwelling as well? This seems important. Of course, there may be differences between reanalyses but you can add, for example, estimates from MERRA-2 and ERA-Interim. This should be easy to do as you can use the TEM residual circulation estimates from the SPARC Reanalysis Intercomparison Project

**-> We now provide the upwelling trends of ERA5 in Figure 2 and Figure 5 and Tab. 2.**

-Page 24, Line 6: It is not clear to me what the discrepancy is here that you are claiming between the GEOSCCM results presented in that study compared to the ones the authors show in Figure 1. Please explain in more detail.

**->As we restructured the text, we exclude that sentence now.**

-The language throughout could be improved at various places. I have noted a few grammatical errors below but there are many others. I strongly encourage the main author to have all co-authors check for lingering language issues/typos.

**-> We revised the language to the best of our capabilities, and took all comments below into account.**

Technical Points:

-Page 1, Line 7: Please indicate a reference for CCMI

**-> Done.**

-Page 2, Line 6: "results from" -> "result from"

**-> Done.**

-Various paragraphs throughout are not indented which renders the formatting a bit awkward (e.g. Page 12, Line 5). Please fix.

**-> Done.**

-Page 12, Line 3: "depending on" -> "dependent on"

**-> Done.**

-Page 12, Line 11: Do you need to remove "not" in front of significant? This is confusing given that the next sentence implies that the trends are significantly related.

**-> Done.**

-Page 15, Line 3: The sentence starting with "We will show. . ." is not complete. -Page 15, Line 15: "evolve" -> do you mean "simulate"?

**-> Done.**

[revised manuscript text omitted]

| | ? | | |
| CESM1-WACCM | ?? | interactive | REF-C2(4)*, fGHG(3), fODS(3) |
| | ? | | |
| EMAC-L90 | ?? | prescribed | REF-C2(1)  |
| EMAC-L47 | ?? | prescribed | REF-C2(1) |
| EMAC-L47-o | ?? | interactive | REF-C2(1)** |
| GEOSCCM | ?? | prescribed | REF-C2(1) |
| | ?? | | |
| MRI | ? | interactive | REF-C2(1) |
| | ?? | | |
| SOCOLv3 | ?? | prescribed | REF-C2(1) |
| NIWA-UKCA | ?? | interactive | REF-C2(5), fGHG(2), fODS(2) |
| | ? | | |
| ULAQ | ? | prescribed | REF-C2(3), fGHG(1), fODS(1) |
| HadGEM | ?? | interactive | REF-C2(1) |
| | ?? | | |
| | ?? | | |
| UMUKCA | ?? | prescribed | REF-C2(2) |
| ACCESS-CCM | ?? | prescribed | REF-C2(3), fGHG(1), fODS(1) |
| | ? | | |
| NIES | ?? | prescribed | REF-C2(1), fGHG(1), fODS(1) |
| UMSLIMCAT | ? | prescribed | REF-C2(1), fGHG(1), fODS(1) |
| CHASER | ? | interactive | REF-C2(1), fGHG(1), fODS(1) |
| LMDz-REPROBUS | ?? | interactive | REF-C2(1) |
| | ? | | |

[revised manuscript text omitted]

**Reply to Anonymous Referee #2** (ACP-2020-947)

We thank referee #2 for the positive and constructive comments on our manuscript. Due to the suggestions of referee #1 we include some additional analysis to this paper. With that we gained additional insight to the possible processes determining the ozone trends in the LS. Due to the new results we also reorganized the structure of the manuscript in the last sections (see new section 3.4 and 3.5). Moreover, note that we do not consider interannual correlations any more, as we felt that not too much was learned here.

Below this reply, you find the revised manuscript considering all questions and comments. Additionally, we highlighted the changes of the manuscript for the pages 1-19, and attached them to the reply. From Section 3.4 on, the changes were rather comprehensive (e.g. replacement of a whole section), that it doesn't make sense to highlight them.

Minor issues:

1) There are a lot of minor editorial issues, especially missing hyphens (e.g. 'free running') and missing commas. I assume that the editorial office will sort these out if the authors don't find them.
**-> We re-read the manuscript carefully and found some of these minor issues. If there are remaining ones, they will surely be corrected during the typesetting process.**

2) Page 1. Line 10-11. It is not clear what the 'systematic change' relates to. The sentence mentions 'different analysis periods', but for the change to be systematic the period would have to be changing in a particular direction?
**-> Rephrased.**

3) Page 2. Line 4. Need to say '1987 Montreal Protocol and later Adjustments/Amendments'. Also, different ODSs started to decline at different times. Some (e.g. HCFCs) might still be growing. You mean the total halogen loading from ODSs.
**-> Done.**

4) Page 2. Line 8. There were still ODSs in atmosphere in 1980. Need to say 'pre-1980' or similar.
**-> Done.**

5) Page 3. Line 12. Put references in chronological order.
**-> Done.**

6) Page 4. Line 14. Ball et al (2018) also included some SD runs which showed very poor agreement with the observed trends. That could be mentioned here.
**-> Done.**

7) Page 5. Footnote. 'fourth'.
**-> Done.**

8) Page 8. Line 22. Please state which model was used by Stone et al. Is that one of the CCMI models?
**-> Done.**

9) Page 9. Caption (and elsewhere). Post 1998 is not the 'post ODS' period. ODSs are still present and different ones have different trends. Total chlorine and bromine are declining, which is not the same thing. You should find another description.

**-> Thank you for this helpful comment, we now do not use the expression post-ODS any more, which was indeed a min-nomer.**

10) Page 12. Line 3. 'depend'.
**-> Done.**

11) Page 12. Line 11. 'low not significant'. This reads strangely. Maybe it is a lack of a comma, but could also be better to say 'non-significant'.
**-> Done.**

[revised manuscript text omitted]

| | ? | | |
| CESM1-WACCM | ?? | interactive | REF-C2(4)*, fGHG(3), fODS(3) |
| | ? | | |
| EMAC-L90 | ?? | prescribed | REF-C2(1)  |
| EMAC-L47 | ?? | prescribed | REF-C2(1) |
| EMAC-L47-o | ?? | interactive | REF-C2(1)** |
| GEOSCCM | ?? | prescribed | REF-C2(1) |
| | ?? | | |
| MRI | ? | interactive | REF-C2(1) |
| | ?? | | |
| SOCOLv3 | ?? | prescribed | REF-C2(1) |
| NIWA-UKCA | ?? | interactive | REF-C2(5), fGHG(2), fODS(2) |
| | ? | | |
| ULAQ | ? | prescribed | REF-C2(3), fGHG(1), fODS(1) |
| HadGEM | ?? | interactive | REF-C2(1) |
| | ?? | | |
| | ?? | | |
| UMUKCA | ?? | prescribed | REF-C2(2) |
| ACCESS-CCM | ?? | prescribed | REF-C2(3), fGHG(1), fODS(1) |
| | ? | | |
| NIES | ?? | prescribed | REF-C2(1), fGHG(1), fODS(1) |
| UMSLIMCAT | ? | prescribed | REF-C2(1), fGHG(1), fODS(1) |
| CHASER | ? | interactive | REF-C2(1), fGHG(1), fODS(1) |
| LMDz-REPROBUS | ?? | interactive | REF-C2(1) |
| | ? | | |

[revised manuscript text omitted]

EMAC-L90 -0.71 height trop. upwelling trend (70 hPa) [-0.82; -0.54 kg/sec/dec]

EMAC-L47 -0.60 trop. upwelling trend (100 hPa) [-0.82; -0.54 kg/sec/dec]

EMAC-L47-o -0.76 downwelling trend (70 hPa) [-0.86; -0.61 kg/sec/dec]

NIES -0.78 downwelling trend (100 hPa) [-0.87; -0.64 kg/sec/dec]

ACCESS -2 -0.54 height trop. ozone trend [-0.71; -0.32 DU/dec]

NIWA -2 -0.37 mid-lat. (fixed) ozone trend [-0.58; -0.11 DU/dec]

NIWA -3 -0.29 mid-lat. (dyn) ozone trend [-0.52; -0.02 DU/dec]

NIWA -4 -0.33 -0.55; -0.08 0.35 0.09; 0.56 -0.64 -0.78; -0.45 NIWA -5 -0.11 [-0.36; 0.17] 0.08 [-0.19; 0.35] -0.37 -0.58; -0.11 ULAQ -1 0.34

**3.5 Forced ozone trends in models**

**4 Discussion**

In the previous sections we analyzed the ozone trends of recent post-ODS periods (i.e. 1998-2018 and variations on this period) the recent 20 year period in detail and found that modeled and observational observed ozone trends disagree, especially in the northern mid-latitude LS. One possible reason for the disagreement between modeled and observed LS ozone trends could be problems with the satellite records, as e.g. instrument biases and drifts can lead to large uncertainties in the observations (see e.g. ???). However, assuming the Assuming the observational data are correct, the question that arises from our results is whether the disagreement stems from the influence of natural variability, or whether it is related to the forced trend. In the forced response to GHG or ODS concentrations is not captured correctly in the models. Thus in the following, we will discuss this question using Fig. 8, which summarizes our results: it shows the observational investigate the relative role of GHG versus ODS forcing on the ozone trends in the models for the observational period and periods extending into the future. Figs. 8a and 8b show upper and lower stratosphere MMM ozone trends in the tropics (20°N-20°S), in the northern mid-latitudes (30°-50°N) and in the southern mid-latitudes (30°-50°S) for the REF-C2 simulations as well as the REF-C2 for the sensitivity simulations with fixed ODS (fODS) and with fixed GHG (fGHG) concentrations (for a detailed description of these sensitivity simulations see Section 2.1). These MMM ozone trends and its spread for periods of different length, for the tropics, the northern mid-latitudes and the southern mid-latitudes in the upper and lower stratosphere are calculated for the recent time period (1998–2018), for a time period, which extends into the future (1998–2040) and for a future time period (2050–2100).